# THRESHOLDED LEXICOGRAPHIC ORDERED MULTI-OBJECTIVE REINFORCEMENT LEARNING

## ABSTRACT

Lexicographic multi-objective problems, which impose a lexicographic importance order over the objectives, arise in many real-life scenarios. Existing Reinforcement Learning work directly addressing lexicographic tasks has been scarce. The few proposed approaches were all noted to be heuristics without theoretical guarantees as the Bellman equation is not applicable to them. Additionally, the practical applicability of these prior approaches also suffers from various issues such as not being able to reach the goal state. While some of these issues have been known before, in this work we investigate further shortcomings, and propose fixes for improving practical performance in many cases. We also present a policy optimization approach using our Lexicographic Projection Optimization (LPO) algorithm that has the potential to address these theoretical and practical concerns. Finally, we demonstrate our proposed algorithms on benchmark problems.

## 1 INTRODUCTION

The need for multi-objective reinforcement learning (MORL) arises in many real-life scenarios and the setting cannot be reduced to single-objective reinforcement learning tasks in general Vamplew et al. (2022). However, solving multiple objectives requires overcoming certain inherent difficulties. In order to compare candidate solutions, we need to incorporate given user preferences with respect to the different objectives. This can lead to *Pareto optimal* or non-inferior solutions forming a set of solutions where no solution is better than another in terms of all objectives. Various methods of specifying user preferences have been proposed and evaluated along three main fronts: (a) expressive power, (b) ease of writing, and (c) the availability of methods for solving problems with such preferences. For example, writing preference specifications that result in a partial order of solutions instead of a total order makes the specification easier for the user but may not be enough to describe a unique preference. Three main motivating scenarios differing on when the user preference becomes available or used have been studied in the literature. (1) User preference is known beforehand and is incorporated into the problem *a priori*. (2) User preference is used *a posteriori*, i.e., firstly a set of representative Pareto optimal solutions is generated, and the user preference is specified over it. (3) An interactive setting where the user preference is specified gradually during the search and the search is guided accordingly.

The most common specification method for the a priori scenario is *linear scalarization* which requires the designer to assign weights to the objectives and take a weighted sum of the objectives, thus making solutions comparable Feinberg & Shwartz (1994). The main benefit of this technique is that it allows the use of many standard off the shelf algorithms as it preserves the additivity of the reward functions. However, expressing user preference with this technique requires significant domain knowledge and preliminary work in most scenarios Li & Czarnecki (2019). While it can be the preferred method when the objectives can be expressed in comparable quantities, e.g. when all objectives have a monetary value, this is not the case most of the time. Usually, the objectives are expressed in incomparable quantities like money, time, and carbon emissions. Additionally, a composite utility combining the various objectives, and an approximation of that with linear scalarization limits us to a subset of the Pareto optimal set.

To address these drawbacks of linear scalarization, several other approaches have been proposed and studied. Nonlinear scalarization methods like Chebyshev Perny & Weng (2010) are more expressive and can capture all of the solutions in the Pareto optimal set, however, they do not address the user-friendliness requirement. In this paper, we will focus on an alternative specification method that overcomes both limitations of linear scalarization, named *Thresholded Lexicographic Ordering*

(TLO) Gábor et al. (1998) Li & Czarnecki (2019). In lexicographic ordering, the user determines an importance order for the objectives, and the less important objectives are only considered if two solutions respect the ordering of the more important objectives. The thresholding part of the technique allows a more generalized definition for being the same w.r.t. an objective. The user provides a threshold for each objective except the last, and the objective values are clipped at the corresponding thresholds. This allows the user to specify values beyond which they are indifferent to the optimization of an objective. There is no threshold for the last objective as it is considered an unconstrained open-ended objective.

Despite the strengths of this specification method, the need for a specialized algorithm to use it in reinforcement learning (RL) has prevented it from being a common technique. The *Thresholded Lexicographic Q-Learning* (TLQ) algorithm was proposed as such an algorithm and has been studied and used in several papers Li & Czarnecki (2019) Hayes et al. (2020). While it has been noted that this algorithm does not enjoy the convergence guarantees of its origin algorithm (Q-Learning), we found that its practical use is limited to an extent that has not been discussed in the literature before. In this work, we investigate such issues of TLQ further. We also present a *Policy Gradient algorithm* as a general solution that has the potential to address many of the shortcomings of TLQ algorithms.

***Our Contributions.*** Our main contributions in this work are as follows: (1) We demonstrate the shortcomings of existing TLQ variants on a common control scenario where the primary objective is reaching a goal state and the other secondary objectives evaluate trajectories taken to the goal. We formulate a taxonomy of the problem space in order to give insights into TLQ's performance in different settings. (2) We propose a *lexicographic projection algorithm* which computes a lexicographically optimal direction that optimizes the current unsatisfied highest importance objective while preserving the values of more important objectives using projections onto hypercones of their gradients. Our algorithm allows adjusting how conservative the new direction is w.r.t. preserved objectives and can be combined with first-order optimization algorithms like Gradient Descent or Adam. We also validate this algorithm on a simple optimization problem from the literature. (3) We explain how this algorithm can be applied to policy-gradient algorithms to solve Lexicographic Markov Decision Processes (LMDPs) and experimentally demonstrate the performance of a REIN-FORCE adaptation on the cases that were problematic for TLQ.

Additionally, in Appendices C and D, we give further insights into TLQ by giving more details about how different TLQ variants fail in problematic scenarios. Then, we present both some of our failed efforts and the promising directions we identified in order to guide future research.

## 2 RELATED WORK

Gábor et al. (1998) was one of the first papers that investigate the use of RL in multi-objective tasks with preference ordering. It introduces TLQ as an RL algorithm to solve such problems. Vamplew et al. (2011) showed that TLQ significantly outperforms Linear Scalarization (LS) when the Pareto front is globally concave or when most of the solutions lie on the concave parts. However, LS performs better when the rewards are not restricted to terminal states, because TLQ cannot account for the already received rewards. Later, Roijers et al. (2013) generalized this analysis by comparing more approaches using a unifying framework. To our knowledge, Vamplew et al. (2011) is the only previous work that explicitly discussed shortcomings of TLQ. However, we found that TLQ has other significant issues that occur even outside of the problematic cases they analyze.

Wray et al. (2015) introduced Lexicographic MDP (LMDP) and the Lexicographic Value Iteration (LVI) algorithm. LMDPs define the thresholds as slack variables which determine how worse than the optimal value is still sufficient. While Wray et al. (2015) proved the convergence to desired policy if slacks are chosen appropriately, such slacks are generally too tight to allow defining user preferences. This is also observed in Pineda et al. (2015) which claimed that while ignoring these slack bounds negates the theoretical guarantees, the resulting algorithm still can be useful in practice.

Li & Czarnecki (2019) investigated the use of Deep TLQ for urban driving. It showed that the TLQ version proposed in Gábor et al. (1998) introduces additional bias which is especially problematic in function approximation settings like deep learning. Also, it depends on learning the true Q function, which can not be guaranteed. To overcome these drawbacks, it used slacks instead of the static thresholds and proposed a different update function. Hayes et al. (2020) used TLQ in a multi-objective multi-agent setting and proposed a dynamic thresholding heuristic to deal with the explosion of the number of thresholds to be set.

However, we discovered that these works on using a Q-learning variant with thresholded ordering perform very poorly in most cases due to non-Markovianity of the value function they try to learn. It is possible to bypass this issue by using policy gradient approaches as they do not require learning an optimal value function. In order to handle conflicting gradients, some modifications to the gradient descent algorithm are needed. Recent work on modified gradient descent algorithms came mostly from Multi Task Learning literature, which could be considered a multiobjective optimization problem Désidéri (2009) Sener & Koltun (2018) Lin et al. (2019) Mahapatra & Rajan (2020) Parisi et al. (2014) Liu et al. (2021). While these papers use similar ideas with our work, their setting is different than ours as they do not have any explicit importance order; hence, not applicable to our setting. Uchibe & Doya (2008) has the most similar setting to ours in gradient-based algorithms. It considers a set of constraints with an unconstrained objective. Then, the gradient of the unconstrained objective is projected onto the positive half-space of the active (violated) constraints and adds a correction step to improve the active constraints. When no valid projection is found, the most violated constraints are ignored until a valid projection exists. This is one of the main differences with our setting: As we have an explicit importance-order of the objectives, it is not acceptable to ignore a constraint without considering the importance order. Also, we project the gradients onto hypercones instead of hyperplanes, which is a hypercone with $\pi/2$ vertex angle. Thus, our algorithm allows varying degrees of conservative projections to prevent a decline in the constraints.

While there are many other recent works on Constrained Markov Decision Process (CMDPs) (Wachi & Sui, 2020; García et al., 2017; Junges et al., 2016), their approaches are not applicable as an importance order over the constraints is not allowed. Recently, using RL with lexicographic ordering began to attract attention from other communities as well. For example, Hahn et al. (2021) uses formal methods to construct single objective MDPs when all of the objectives are $\omega$-regular.

Finally, Skalse et al. (2022) was published in August 2022 and it proposes both value-based and policy-based approaches. Their value-based approach is based on slacks like Li & Czarnecki (2019) and they require using very small slack values. This protects their approach from the issues with having relaxations by limiting their setting to strict lexicographic order. For policy-based methods, they use Lagrangian relaxation and their setting is again a strict lexicographic ordering, i.e. it does not allow treating values above a threshold equal.

## 3 BACKGROUND

**Multiobjective Markov Decision Process (MOMDP).** A MOMDP is a tuple $\langle S, A, T, \mathbf{R}, \gamma \rangle$ where

- $S$ is the finite set of states with initial state $s_{init} \in S$ and a set of terminal states $S_F$,
- $A$ is a finite set of actions,
- $P$: $S \times A \times S \to [0, 1]$ is the state transition function given by $P(s, a, s') = \mathbb{P}(s'|s, a)$, the probability of transitioning to state $s'$ given current state $s$ and action $a$.
- $\mathbf{R} = [R_1, \dots, R_K]^T$ is a vector that specifies the reward of transitioning from state $s$ to $s'$ upon taking action $a$ under $K$ different reward functions $R_i : S \times A \times S \to \mathbb{R}$ for $i \in \{1, \dots, K\}$.
- $\gamma \in \mathbb{R}$ is a discount factor.

In such a MOMDP, a finite *trajectory* $\zeta \in (S \times A)^* \times S$ is a sequence $\zeta = s_0 a_0 s_1 a_1 \dots a_{T-1} s_T$ where $s_i \in S$, $a_i \in A$ and indices denote the time steps. The evolution of an MDP is governed by repeated agent-environment interactions, where in each step, an *agent* first picks actions in a state $s$ according to some probabilistic distribution, and for each of these actions $a$ the *environment* generates next states according to $\mathbb{P}(s'|s, a)$. Each reward function $R_i$ corresponds to an *objective* $o_i$, the discounted rewards sum that the agent tries to maximize. Control over a MOMDP requires finding an optimal *policy* function $\pi^* : S \times A \to [0, 1]$ which assigns probability $\mathbb{P}_{\pi^*}(a|s)$ to actions $a \in A$. In this paper, we use the *episodic* case of MDP where the agent-environment interaction consists of sequences that start in $s_{init}$ and terminates when a state in $S_F$ is visited. The length of the episode is finite but not a fixed number. In MDP literature, this is known as "indefinite-horizon" MDP. The episodic case can be ensured by restricting ourselves to suitable policies which have a non-zero probability for all action in all states.

We define the quality of a policy $\pi$ with respect to an objective $o_i \in \{1, \dots, K\}$ by the value function $V_i^\pi : S \to \mathbb{R}$ given by $V_i^\pi(s) = \mathbb{E}_\pi[\sum_{t=0}^T \gamma^t R_i(s_t, a_t, s_{t+1})|s_0 = s]$. Intuitively, $v_i^\pi(s)$ is the expected return from following policy $\pi$ starting from state $s$ w.r.t. objective $o_i$. Overall, the *quality* of a policy $\pi$ is given by the vector valued function $\boldsymbol{V^\pi} : S \to \mathbb{R}^K$ which is defined as

$\boldsymbol{V^{\pi}}(s) = [V_1^{\pi}(s), \ldots, V_K^{\pi}(s)]^T$. As $\boldsymbol{V}$ is vector-valued, without a preference for comparing $V_i^{\pi}$ values across different $i$, we only have a partial order over the range of $\boldsymbol{V}$, leading to *Pareto front* of equally good quality vectors. Further preference specification is needed to order the points on the Pareto front. A *Lexicographic MDP (LMDP)* is a class of MOMDP which provides such an ordering. It adds another component to MOMDP definition:

- $\tau = \langle \tau_1, \ldots, \tau_{K-1} \rangle \in \mathbb{R}^{K-1}$ is a tuple of threshold values where $\tau_i$ indicates the minimum acceptable value for objective $i$. The last objective does not require a threshold; hence, there are only $K - 1$ values. Then, $\boldsymbol{\tau}$ can be used to compare value vectors $\boldsymbol{u}, \boldsymbol{v} \in \mathbb{R}^K$ by defining the thresholded lexicographic comparison $>^{\boldsymbol{\tau}}$ as $\mathbf{u} >^{\boldsymbol{\tau}} \mathbf{v}$ iff there exists $i \leq K$ such that:
  - $\forall j < i$ we have $\mathbf{u_j} \geq \min(\mathbf{v_j}, \tau_j)$; and
    * if $i < K$ then $\min(\mathbf{u_i}, \tau_i) > \min(\mathbf{v_i}, \tau_i)$,
    * otherwise if $i = K$ then $\mathbf{u_i} > \mathbf{v_i}$.

  Intuitively, we compare $\boldsymbol{u}$ and $\boldsymbol{v}$ starting from the most important objective ($j = 1$); the less important objectives are considered only if the order of higher priority objectives is respected. The relation $\geq^{\boldsymbol{\tau}}$ is defined as $>^{\boldsymbol{\tau}}$ OR $=$.

**Value-function Algorithms for Optimal Policies.** An optimal policy $\pi^*$ is a policy that is better than or equal to any other policy $\pi \in \Pi$, i.e., if $V^{\pi^*}(s) \geq^{\boldsymbol{\tau}} V^{\pi}(s) \, \forall s \in S$ for all other policies $\pi$ (Gábor et al., 1998). There are two approaches to finding an optimal policy in RL: Value-function algorithms and Policy Gradient algorithms. Value function based methods estimate the optimal action-value function $Q^*$ and construct $\pi^*$ using it. The action-value function under $\pi$, $Q^{\pi} : S \times A \to \mathbb{R}^K$, is defined as: $Q^{\pi}(s, a) \triangleq \mathbb{E}_{\pi}[\sum_{t=0}^{T} \gamma^t \mathbf{R}(s_t, a_t, s_{t+1}) | s_0 = s, a_0 = a]$ The optimal action-value function, $Q^{\star}$, is defined as: $Q^{\star}(s, a) = \max_{\pi \in \Pi} Q^{\pi}(s, a)$. Then, $\pi^*$ can be obtained as: $\pi^*(s, a) = 1$ if $a = \arg\max_{a' \in A} Q^{\star}(s, a')$, and 0 otherwise. In single objective MDPs, the Bellman Optimality Equation as seen in Eq. 1 is used to learn $Q^{\star}$ as it gives an update rule that converges to $Q^{\star}$ when applied iteratively.

$$Q^{\star}(s, a) = \mathbb{E}_{s' \sim P}[(R(s, a, s') + \gamma \max_{a' \in A} Q^{\star}(s', a'))] \tag{1}$$

Q-learning Watkins & Dayan (1992) is a very popular algorithm that takes this approach. TLQ tries to extend Q-learning for LMPDs; however, Bellman Optimality Equation does not hold in LMDPs. Hence, this approach lacks the theoretical guarantees enjoyed by Q-learning.

**Policy Gradient Algorithms for $\pi^*$.** Policy gradient algorithms in RL try to learn the policy directly instead of inferring it from the value functions. These methods estimate the gradient of the optimality measure w.r.t. policy and update the candidate policy using this potentially imperfect information. We denote the policy parameterized by a vector of variables, $\theta$ as $\pi_{\theta}$. The performance of the policy $\pi_{\theta}$, denoted $J(\theta)$, can be defined as the the expected return from following $\pi_{\theta}$ starting from $s_{init}$, i.e. $J(\theta) \triangleq V^{\pi_{\theta}}(s_{init})$. Once the gradient of the optimality measure w.r.t. the parameters of the policy function is estimated, we can use first-order optimization techniques like Gradient Ascent to maximize the optimality measure. While all based on the similar theoretical results, a myriad of policy gradient algorithms have been proposed in the literature Sutton et al. (1999) Konda & Tsitsiklis (1999) Schulman et al. (2017) Lillicrap et al. (2015) Haarnoja et al. (2018).

## 4    TLQ: Value Function Based Approaches for TLO

Previous efforts to find solutions to the LMDPs have been focused on value-function methods. Apart from Wray et al. (2015), which takes a dynamic programming approach, these have been variants of Thresholded Lexicographic Q-Learning (TLQ), an LMDP adaptation of Q-learning Gábor et al. (1998) Li & Czarnecki (2019). While these methods have been used and investigated in numerous papers, the extent of their weaknesses has not been discussed explicitly.

In order to adapt Q-learning to work with LMDPs, one cannot simply use the update rule in Q-learning for each objective and learn the optimal value function of each objective completely independent of the others. Such an approach would result in the actions being suboptimal for some of the objectives. Based on this observation, two variants of TLQ Gábor et al. (1998)Li & Czarnecki (2019) have been proposed, which differ in how they take the other objectives into account. We analyze these variants by dividing their approaches into two components: (1) value functions and update rules; and (2) acceptable policies for action selection. However, it should be noted that these

components are inherently intertwined due to the nature of the problem — the value functions and acceptable policies are defined recursively where each of them uses the other's last level of recursion. Due to these inherent circular referencing, the components will not be completely independent and some combinations of introduced techniques may not work.

**Value Functions and Update Rules.** The starting point of learning the action-value function for both variants is $Q^\star = \langle Q_1^\star, \dots, Q_K^\star \rangle$ where each $Q_i^\star : S \times A \to \mathbb{R}$ is defined as in Section 3 only with the change that the maximization is not done over the set of all policies $\Pi$ but over a subset of it $\Pi_{i-1} \subseteq \Pi$ which will be described below. Gábor et al. (1998) proposes learning $\hat{Q}^\star : S \times A \to \mathbb{R}^K$ where each component of $\hat{Q}^\star$ denoted by $\hat{Q}_i^\star$ is defined as: $\hat{Q}_i^\star(s,a) \triangleq \min(\tau_i, Q_i^\star(s,a))$ In other words, it is the rectified version of $Q_i^\star$. It is proposed to be learned by updating $\hat{Q}_i^\star(s,a)$ with the following value iteration which is adapted from Eq. 1

$$\min\left(\tau_i, \sum_{s'} P(s,a,s')(R_i(s,a,s') + \gamma \max_{\pi \in \Pi_{i-1}} \hat{Q}_i^\star(s', \pi(s')))\right) \qquad (2)$$

Notice that similar to the definition of $Q_i^\star$, the main change during the adaptation is limiting the domain of max operator to $\Pi_{i-1}$ from $\Pi$. On the other hand, Li & Czarnecki (2019) proposes that we estimate $Q^\star$ instead and use it when the actions are being picked. This $Q^\star$ uses the same update rule as Eq. 1 with only change being maximization over $\Pi_{i-1}$.

**Acceptable Policies $\Pi_i$ and Action Selection.** The second important part of TLQ is the definition of "Acceptable Policies", $\Pi_i$, which is likely to be different for each objective. The policies in $\Pi_i$ are ones that satisfy the requirements of the first $i$ objectives. Values of the acceptable policies in a given state are the acceptable actions in that state. Hence, these sets will be used as the domain of both max operator in the update rules and arg max operator in `ActionSelection` function. The pseudocode of this function can be seen in Algorithm 1. Note that the structure of this function is the same for both variants of TLQ and different instantiations of the function differ in how the `AcceptableActs` subroutine is implemented. `AcceptableActs` takes the current state $s$, the Q-function to be used, and the actions acceptable to the objectives up to the last one and outputs the actions acceptable to the objectives up to and including the current one. Below, we will describe how $\Pi_i$ has been defined in the literature, see Appendix B for the formal definitions.

---

**Algorithm 1** ActionSelection

**Function** `ActionSelection`$(s, Q, \epsilon | A)$**:**
  $r \sim U(0,1)$
  **if** $r < \epsilon$ **then**
    $a$ is picked randomly from $A$
  **else**
    $A_0 \leftarrow A$
    **for** $o = 1, K$ **do**
      **if** $|A_{o-1}| > 1$ **and** $o < K$ **then**
        $A_o \leftarrow$ `AcceptableActs`$(s, Q, A_{o-1})$
      **else**
        $a \leftarrow \arg\max_{a' \in A_{o-1}} Q_i(s, a')$
        **break**
    **end**
  **return** $a$

---

*Absolute Thresholding:* Gábor et al. (1998) proposes this approach where the actions with values higher than a real number are considered acceptable. Hence, $\Pi_i$ is the subset of $\Pi_{i-1}$ for which Q-values are higher than some $\tau$.

*Absolute Slacking:* This is the approach taken by Li & Czarnecki (2019) where a slack from the optimal value in that state is determined and each action within that slack is considered acceptable.

### 4.1 SHORTCOMINGS OF PRIOR TLQ APPROACHES

The shortcomings of TLQ depend on the type of task. We introduce a taxonomy to facilitate the discussion.

*Constrained/Unconstrained Objective:* Constrained objectives are bounded in quality by their threshold values above which all values are considered the same. Unconstrained objectives do not have a such threshold value. In an LMDP setting, all objectives but the last one are constrained objectives.
*Terminating Objective:* An objective that either implicitly or explicitly pushes the agent to go to a terminal state of the MDP. More formally, this means discounted cumulative reward for an episode that does not terminate within the horizon is lower than an episode that terminates.
*Reachability Objective:* A reachability objective is represented by a non-zero reward in the terminal states of the MDP and zero rewards elsewhere. We will call an objective that has non-zero rewards in at least one non-terminal state a non-reachability objective.

This taxonomy also helps seeing the problematic cases and potential solutions. To summarize the empirical demonstrations about the applicability of TLQ for different parts of the problem space:

(I) *TLQ successful case studies*: Vamplew et al. (2011) shows an experiment where the constrained objective is reachability, unconstrained objective is terminating, and TLQ works. Li & Czarnecki (2019) shows this for a case study where the constrained objective is a non-terminating reachability constraint. Note that these are just demonstrations of empirical performance for some case studies that fall into these categories and the papers do not make claims about the general instances.

(II) *TLQ does not work*: Vamplew et al. (2011) shows that TLQ does not work when constrained objective is non-reachability. In this work, we show that TLQ also does not work when the constrained objective is a terminating reachability objective but the unconstrained one is non-terminating.

**Benchmark.** We describe some scenarios that are common in control tasks yet TLQ fails to solve. To illustrate different issues caused by TLQ, we need an adaptable multiobjective task. Also, limiting ourselves to finite state and action spaces where the tabular version of TLQ can be used simplifies the analysis. Our MAZE environment satisfies all of our requirements. Figure 1 shows an example.

```
 _____________
|___|G_|___|  2
|HH|HH|___|  1
|___|S_|___|  0
  0   1   2
```

Figure 1: A simple maze that can be used to demonstrate how TLQ fails to reach the goal state.

In all MAZE instantiations, each episode starts in $S$ and $G$ is the terminal state. There are also two types of bad tiles. The tiles marked with $HH$s are high penalty whereas the ones with $hh$s show the low penalty ones. In this work, we will use $-5$ as high penalty and $-4$ as low penalty. But we consider the penalty amounts parameters in the design of a maze; so, they could change. There are two high-level objectives: Reach $G$ and avoid bad tiles. Ordering of these objectives and exact definition of them results in different tasks. We use these different tasks to cover different parts of the problem space described in the taxonomy. The action space consists of four actions: $Up, Down, Left, Right$. These actions move the agent in the maze and any action which would cause the agent to leave the grid will leave its position unchanged.

**Problems with Reachability Constraint.** A common scenario in control tasks is having a primary objective to reach a goal state and a secondary objective that evaluates the path taken to there. Formally, this is a scenario where the primary objective is a reachability objective. However, TLQ either needs to ignore the secondary objective or fails to satisfy the primary objective in such cases. All of the thresholding methods above fail to guarantee to reach the goal in this setting when used threshold/slack values are uniform throughout state space. The maze in Figure 1 can be used to observe this phenomenon. Assume that our primary objective is to reach $G$ and we encode this with a reward function that is $0$ everywhere but $G$ where it is $R$. And our secondary objective is to avoid the bad tiles. A Pareto optimal policy in this maze could be indicated by the state trajectory $(1, 0) \rightarrow (2, 0) \rightarrow (2, 1) \rightarrow (2, 2) \rightarrow (1, 2)$. However, this is unattainable by TLQ.

Since the reward is given only in the goal state, $\hat{Q}_1^\star$ can be equal to $\tau_1$ only for the state-action pairs that lead to the goal state. All others will be discounted from these; hence, all are less than $\tau_1$. This means the agent will always ignore the secondary when using absolute thresholding of Gábor et al. (1998). If no discounting was used, all actions would have the value $\tau_1$, hence the agent would not need to reach the goal state. We believe the reason why Vamplew et al. (2011) has not observed this issue in their experiments with undiscounted ($\gamma = 1$) Deep Sea Treasure (DST) is due to their objectives. The secondary objective of DST, minimizing the time steps before terminal state, is a terminating objective which pushes the agent to actually reach a goal state.

In Appendix C.1, we show how Absolute Slacking also leads to contradicting requirements when the trajectory described above is aimed. Moreover, in Appendix C.2, we present another case study which shows all TLQ variants fail to find the optimal policies in some very standard scenarios where the primary objective is a reachability objective. More generally, it highlights the shortcomings of having a single threshold/slack value for all states which can manifest itself in other settings too.

## 5 POLICY GRADIENT APPROACH FOR TLO

In this section, we introduce our policy gradient approach that utilizes consecutive gradient projections to solve LMDPs. Policy gradient methods treat the performance of a policy, $J(\theta)$, as a function of policy parameters that needs to be maximized and employ standard gradient ascent optimization algorithms Ruder (2016). Following this modularity, we start with proposing a general optimization algorithm, the Lexicographic Projection Algorithm (LPA), for multiobjective optimization (MOO)

problems with thresholded lexicographic objectives. Then, we will show how single objective Policy Gradient algorithms can be adapted to this optimization algorithm.

As gradients w.r.t. different objectives can be conflicting for MOO, various ways to combine them have been proposed. Uchibe & Doya (2008) proposes projecting the gradients of the less important objectives onto the positive halfspaces of the more important gradients. Such a projection vector has the highest directional derivative w.r.t. less important objective among the directions with non-negative derivative w.r.t. the important objectives. This is assumed to protect the current level for the important objective while trying to improve the less important. However, non-negative derivative actually does not guarantee to protect the current level as infinitely small step sizes are not used in practice. For example, if a function is strictly concave, the change in a zero-derivative direction will be always negative for any step size greater than $0$. Therefore, we propose projecting onto *hypercones* which allows more control over *how safe* the projection is. A hypercone is the set of vectors that make at most $\frac{\pi}{2} - \Delta$ angle with the axis vector; a positive halfspace is a special hypercone where $\Delta = 0$. Increasing $\Delta$ brings the projection closer to the axis vector. A hypercone $C_a^\Delta$ with axis $a \in \mathbb{R}^n$ and angle $\frac{\pi}{2} - \Delta$ is defined as $C_a^\Delta =$

$$\left\{ \boldsymbol{x} \in \mathbb{R}^n \left| \|\boldsymbol{x}\| = 0 \vee \frac{\boldsymbol{a}^T \boldsymbol{x}}{\|\boldsymbol{a}\| \|\boldsymbol{x}\|} \geq \cos\left(\frac{\pi}{2} - \Delta\right) \right. \right\} \tag{3}$$

We can derive the equation for projection of $\boldsymbol{g}$ on $C_a^\Delta$ by firstly showing that $\boldsymbol{g}$, $\boldsymbol{a}$, and the projection $\boldsymbol{g^p}$ are planar using Karush-Kuhn-Tucker (KKT) conditions. Then, we can derive the formula by using two-dimensional geometry (the details are in the appendix); giving us $\boldsymbol{g^p} =$

$$\frac{\cos \Delta}{\sin \phi} \sin (\Delta + \phi) \left( \boldsymbol{g} + \boldsymbol{a} \frac{\|\boldsymbol{g}\|}{\|\boldsymbol{a}\|} (\sin \phi \tan \Delta - \cos \phi) \right) \tag{4}$$

where $\phi$ is the angle between $\boldsymbol{a}$ and $\boldsymbol{g}$. Moving forward, we assume a function $projectCone(\boldsymbol{g}, \boldsymbol{a}, \Delta)$ which returns $\boldsymbol{g^p}$ according to this equation.

**Lexicographic Projection Algorithm (LPA).** A *Thresholded Lexicographic MOO (TLMOO)* problem with $K$ objectives and $n$ parameters can be formulated as maximizing a function $F : A \to \mathbb{R}^K$ where $A \in \mathbb{R}^n$, and the ordering between two function values $F(\theta_1), F(\theta_2)$ is according to $\geq^\tau$ as defined as in Section 3. Notice that when we have multiple objectives, the gradients will form a $K$-tuple, $G = (\nabla F_1, \nabla F_2, \cdots, \nabla F_K)$, where $\nabla F_i$ is the gradient of $i^{th}$ component of $F$.

Since TLMOO problems impose a strict importance order on the objectives and it is not known how many objectives can be satisfied simultaneously beforehand, a natural approach is to optimize the objectives one-by-one until they reach the threshold values. However, once an objective is satisfied, optimizing the next objective could have a detrimental effect on the satisfied objective. We can use hypercone projection to avoid this. More formally, when optimizing $F_i$, we can project $\nabla F_i$ onto the intersection of $\{C_{\nabla F_j}^\Delta\}_{j<i}$ where $\Delta$ is a hyperparameter representing how risk-averse our projection is, and use the projection as the new direction. If such an intersection does not exist, it means that it is not possible to improve $F_i$ without taking a greater risk and we can terminate the algorithm.

Improving this approach with a heuristic that prevents overly conservative solutions leads to better performance in certain cases. Conservative updates usually lead to further increases on the already satisfied objectives instead of keeping them at the same level. This means most of the time, we have enough buffer between the current value of the satisfied objectives and their thresholds to sacrifice some of it for further gains in the currently optimized objective. Then, we can define a set of "active constraints" which is a subset of all satisfied objectives that we will not accept any sacrifice and only consider these when projecting the gradient. The "active constraints" can be defined loosely, potentially allowing a hyperparameter that determines the minimum buffer zone needed to sacrifice from an objective.

The `FindDirection` function in Algorithm 2 (our LPA algorithm) incorporates these ideas. This function takes the tuple of all gradients $M$, the tuple of current function values $F(\theta)$, threshold values $\tau$, the conservativeness hyperparameter $\Delta$, a boolean $AC$ that determines whether "active constraints" heuristic will be used or not, and a buffer value $b$ to be used alongside active constraints heuristic as inputs. Then, it outputs the direction that should be followed at this step, which can replace the gradient in a gradient ascent algorithm. For the optimization experiments, we will be using the vanilla gradient ascent algorithm. Algorithm 2 finds the first objective that has not passed its threshold and iteratively projects its gradient onto hypercones of all previous objectives. If such

---

**Algorithm 2** Lexicographic Constrained Ascent Direction

---

**Function** `FindDirection`$(M, F(\theta), \tau, \Delta, AC, b)$**:**

 Initialize action-value function $Q$ with random weights

 **for** $o = 1, K$ **do**

  **if** $o = K - 1$ **or** $F_o(\theta) < \tau_o$ **then**

   Initialize direction $\boldsymbol{u}$ with initial state $M_o$

   **for** $j = 1, o$ **do**

    **if not** *((AC* **and** $F_j(\theta) > \tau_j + b$*) or* $\angle(\boldsymbol{u}, M_j) < \frac{\pi}{2} - \Delta$*)* **then**

     $\boldsymbol{u} \leftarrow projectCone(\boldsymbol{u}, M_j, \Delta)$

   **end**

   **for** $j = 1, o + 1$ **do**

    **if not** *((j $\neq$ K − 1* **and** *AC* **and** $F_j(\theta) > \tau_j + b$*) or* $\angle(\boldsymbol{u}, M_j) < \frac{\pi}{2} - \Delta$*)* **then**

     **return None**

   **end**

   **return** $\boldsymbol{u}$

 **end**

---

a projection exists, it returns the projection as the "Lexicographic Constrained Ascent" direction. Otherwise, it returns null. In our experiments, we will set $b = 0$. In general, $b$ can be set to any non-negative value and higher values of $b$ would result in a more conservative algorithm which does not sacrifice from an objective unless it is *well* above the threshold.

**Experiment.** As a benchmark for the Lexicographic Projection Algorithm, we used $F_1(x, y) = -4x^2 + -y^2 + xy$ and $F_2(x, y) = -(x - 1)^2 - (y - 0.5)^2$ which are taken from Zerbinati et al. (2011). We modified $F_2$ slightly for better visualization and multiplied both functions with $-1$ to convert this to a maximization problem. We set the threshold for $F_1$ to $-0.5$. The behavior of our cone algorithm without using active constraints heuristic on this problem with $\tau = (-0.5)$ and $\Delta = \frac{\pi}{90}$ can be seen in Figure 2. Further results with $AC$ heuristic can be found in Appendix F.4.

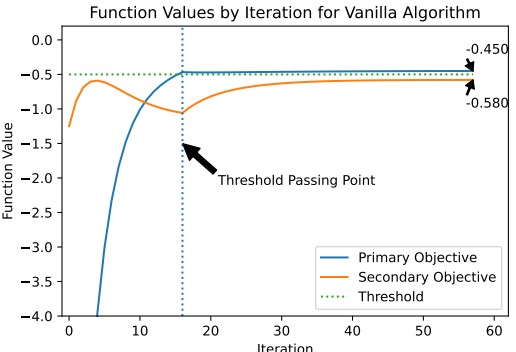

**Using Lexicographic Projection Algorithm in RL.** We show how LPA can be combined with policy gradient algorithms. We use REINFORCESutton et al. (1999) as the base policy gradient algorithm because its simplicity minimizes conceptual overhead.

We can adapt REINFORCE to work in LMDPs by repeating the gradient computation for each objective independently and computing a new direction using `FindDirection` function. Then, this new direction can be passed to the optimizer. Algorithm 5 in the appendix shows the pseudocode for this algorithm.

Figure 2: The changes in the function values. Notice that $F_2$, in orange, is completely ignored until the threshold for $F_1$ is reached. Then, the algorithm optimizes $F_2$ while respecting the passed threshold of $F_1$.

Note that our algorithm is compatible with most policy gradient algorithms. Uchibe & Doya (2008) shows how a similar idea is applied to actor-critic family of policy gradient algorithms which reduces the variance in the gradient estimation by using a *critic* network. We believe that more stable policy gradient algorithms like actor-critic methods could further improve the performance of lexicographic projection approach as our algorithm might be sensitive to noise in gradient estimation.

**Experiments.** We evaluate the performance of the Lexicographic REINFORCE algorithm on two Maze problems. In both experiments, we use a two layer neural network (LeCun et al., 2015) for policy function. Details of the policy function can be found in Appendix G.3.

**Reachability Experiment.** As the first experiment, we consider the case where the primary objective is a reachability objective and the secondary objective is non-terminating, which was the setting that we found that TLQ fails to reach the goal state in Section 4.1. Our experiments show that Lexicographic REINFORCE can successfully solve this problem. Details of this experiment and results can be found in Appendix G.3.2.

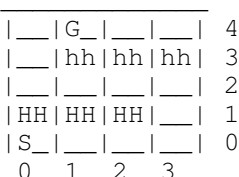

```
 ___________________
|___|G_|___|___|  4
|___|hh|hh|hh|    3
|___|___|___|___|  2
|HH|HH|HH|___|    1
|S_|___|___|___|  0
  0   1   2   3
```

Figure 3: The Maze for the non-reachability experiment.

**Non-reachability Experiment.** The primary objective is a non-reachability objective, i.e. it takes non-zero values in some non-terminal states. For this, we flip our objectives from the previous setting and define our primary objective as minimizing the cost incurred from the bad tiles. $HH$s give $-5$ reward and $hh$s give $-4$ reward. A $+1$ reward is given in the terminal state to extend the primary objective to have rewards in both terminal and non-terminal states. The secondary objective is to minimize the time taken to the terminal state. We formalize this by defining our secondary reward function as $0$ in terminal state and $-1$ everywhere else. We use the Maze in Figure 3 for this experiment. Note that this is the setting that Vamplew et al. (2011) has found that TLQ fails. However, our experiments show that Lexicographic REINFORCE can solve this setting too.

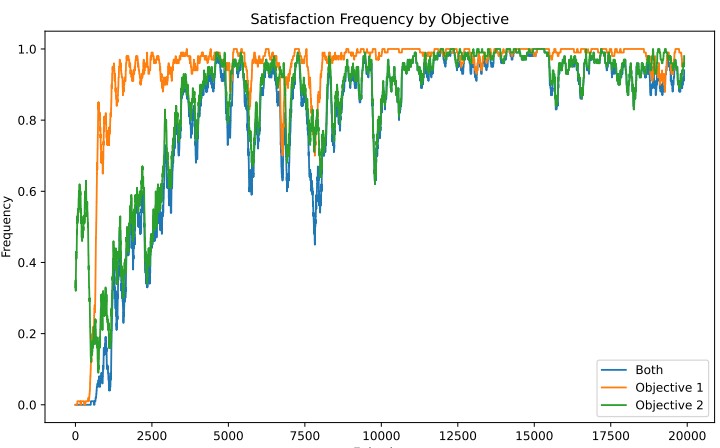

Figure 4: Satisfaction frequency for a single successful seed for the non-reachability experiment.

We found that out of the 10 seeds, 7 find policies that have 90% success over 100 episodes. The change in satisfaction frequencies of individual objectives for a successful seed can be seen in Figure 4. Notice that the primary objective initially starts very low and quickly increases while the secondary objective does the opposite. This happens while the algorithm mostly optimizes the primary objective. Once the primary objective is learned, the algorithm starts improving the secondary objective.

These experiments illustrate the usefulness of projection based policy gradient algorithm for different tasks. We believe that these results can be generalized to more complex tasks when our algorithm is combined with more stable and sophisticated policy gradient algorithms.

## 6 CONCLUSION

In this work, we considered the problem of solving LMDPs using model-free RL. While previous efforts on this problem have been focused on value-function based approaches, the applicability of these approaches over different problem settings and investigation of the extent of their shortcomings have been limited. Our first contribution was providing further insights into the inherent difficulties of developing value function based solutions to LMDPs. Towards this end, we both illustrated failure scenarios for the existing methods, and also presented (in Appendices C and D) potential new value-function based approaches. These approaches include both our failed attempts and promising directions, we believe both will be helpful for future research.

Our second focus in this work was developing and presenting a policy-gradient based approach for LMDPs. Policy gradient algorithms have not been studied in MDPs with thresholded lexicographic objectives before even though they are more suitable for this setting as they bypass many inherent issues with value functions, such as non-convergence due to non-greedy policies w.r.t. value functions, and the need for different threshold values across the state space. For this, we developed a general thresholded lexicographic multi-objective optimization procedure based on gradient hyper-cone projections. Then, we showed how policy gradient algorithms can be adapted to work with LMDPs using our procedure, and demonstrated the performance of our REINFORCE adaptation. While our results are promising for the REINFORCE adaptation, future research could be further empirical studies with more stable policy-gradient algorithm adaptations, and over more complex tasks.

## REPRODUCIBILITY STATEMENT

In this work, we list all the hyperparameters that we used in the experiments in Section 5 and Appendix G.3. We also share the source code for Section 5 in the supplementary material and we will make the rest of the source code public for the camera-ready version. Moreover, we provide the derivations of the projection formula we use for policy gradient approach, in Appendix E.

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

## A    TECHNICAL APPENDIX ORGANIZATION

In this section, we will give an overview of the Appendix.

*Technical Appendix for Section 4:* In Appendix B, we share further mathematical details for the different acceptable policy definitions. In Appendix C, we firstly describe how Absolute Slacking and Relative Slacking suffer from the failure to reach the goal. Then, we present a different issue that was not covered in the main paper: failure to sacrifice at both ends of the episode. After the discussion of issues, we propose different approaches that can address these issues in Appendix D.

*Technical Appendix for Section 5:* We start by describing the derivation of the projection formula used in the paper in Appendix E. Then, in Appendix F, we show why cone projection is useful and share the rest of the experiments for the LPA algorithm on the simple optimization benchmark used in the main paper too. Finally, we share further results from the experiments with the adapted REINFORCE algorithm in Appendix G.

Within this technical appendix, we would like to highlight the following sections.

Appendix C continues our discussion in Section 4.1 and demonstrates how existing TLQ variants fail in the given scenarios. Appendix D.2 presents a TLQ fix that addresses some shortcomings of TLQ for two objectives. Appendix D.3.3 formulates how the well-known state augmentation idea can be used to solve LMDPs and proposes this as a new research direction.

For the policy gradient part of our work, Appendix F.3 illustrates why hypercone projection is needed instead of halfspace projections for lexicographic optimization. Then, Appendix G.2 shows the pseudocode of our REINFORCE adaptation. Finally, Appendices F.4 and G.3 present rest of the experiments that we could not fit in Section 5.

## B    FURTHER DETAILS ON ACCEPTABLE POLICIES

In this section, we will give the mathematical definitions of different thresholding methods. Also, we will describe Relative Slacking, an alternative thresholding method that does not exist in the literature. We include it for the sake of completeness.

1. **Absolute Thresholding:** This is the approach proposed by Gábor et al. (1998) where the actions with values higher than a real number are considered acceptable. Formally,

$$\Pi_i \triangleq \{\pi_i \in \Pi_{i-1} \mid \hat{Q}_i^\star(s, \pi_i(s)) = \max_{a \in \{\pi_{i-1}(s)|\pi_{i-1} \in \Pi_{i-1}\}} \hat{Q}_i^\star(s, a), \forall s \in \mathcal{S}\} \quad (5)$$

2. **Absolute Slacking:** This is the approach taken by Li & Czarnecki (2019) and Skalse et al. (2022) where a slack from the optimal value in that state is determined and each action within that slack is considered acceptable.

$$\Pi_i \triangleq \{\pi_i \in \Pi_{i-1} \mid Q_i^\star(s, \pi_i(s)) \geq \max_{a \in \{\pi_{i-1}(s)|\pi_{i-1} \in \Pi_{i-1}\}} Q_i^\star(s, a) - \delta_i, \forall s \in \mathcal{S}\} \quad (6)$$

Notice that this thresholding scheme is not directly compatible with our definition of LMDPs in Section 3. While they are both used to simply introduce some relaxation in policy selection and it does not affect our general analysis, see Wray et al. (2015) and Pineda et al. (2015) for a definition based on slacks.

3. **Relative Slacking:** In this approach, slacks are defined as ratios $\eta \in (0, 1]$ rather than absolute values. Then, any action with value greater than $(1 - \eta)$ times the optimal value is considered acceptable. Formally,

$$\Pi_i \triangleq \{\pi_i \in \Pi_{i-1} \mid Q_i^\star(s, \pi_i(s)) \geq (1 - \eta) \max_{a \in \{\pi_{i-1}(s)|\pi_{i-1} \in \Pi_{i-1}\}} Q_i^\star(s, a), \forall s \in \mathcal{S}\} \quad (7)$$

While has not been proposed in any previous work, we included this for the sake of completeness. Notice that "Relative Thresholding" would be essentially the same technique, only with different parameters.

## C    ISSUES WITH TLQ

In this section, we will elaborate more on the different issues with TLQ.

### C.1 Failing to Reach the Goal

In this section, we will explain how Relative Slacking and Absolute Slacking fail to reach the goal, an issue we discussed in Section 4 for Absolute Thresholding. We will again use Figure 1.

Relative Slacking determines the maximum detour. If a non-optimal action delays reaching the goal for $k$ steps, this can be allowed only by defining $\eta > \gamma^k$. However, this detour can be taken repeatedly, preventing actually reaching the goal.

Seeing how Absolute Slacking fails to overcome this problem is a little trickier. It requires a closer inspection of action-values. Since we want that the agent to go left in $(2, 2)$, the following should be true:

$$Q_1^\star((2,2), Right) < Q_1^\star((2,2), Left) - \delta_1$$
$$\implies \gamma R < R - \delta_1$$
$$\implies \delta_1 < R(1 - \gamma)$$

However, allowing the agent to pick $Right$, instead of $Up$ in $(1, 0)$ requires:

$$Q_1^\star((1,0), Right) \geq Q_1^\star((2,2), Up) - \delta_1$$
$$\implies \gamma^3 R \geq \gamma R - \delta_1$$
$$\implies \delta_1 \geq R\gamma(1 - \gamma^2)$$

Combining these two requirements implies that:

$$R(1 - \gamma) > R\gamma(1 - \gamma^2)$$
$$\implies 1 > \gamma(1 + \gamma)$$
$$\implies 0 > \gamma^2 + \gamma - 1 \qquad \text{Solving the quadratic equation}$$
$$\implies 0.62 > \gamma$$

This shows that to reach the desired policy, not only $\delta$ but $\gamma$ needs to be adjusted too. However, the $\gamma$ parameter is assumed to be an environment constant and traditionally set to values close to 1. Moreover, there is no real way to find the correct $\gamma$ value apart from computing the action-value function, the very thing we are trying to compute.

Also, a similar analysis shows that small tricks like replacing the primary reward function with

$$R_1'(s, a, s') = \begin{cases} 0, \text{if } s' = G \\ -1, \text{otherise} \end{cases} \tag{8}$$

with or without discounting does not solve this problem.

### C.2 Failure to Sacrifice Early and Late

In this section, we will discuss an issue that was not discussed in the main paper: the failure to sacrifice in the early and late parts of the episode. This issue still occurs even if "failure to reach the goal" issue is avoided because the secondary objective happened to be a terminating one.

Consider the maze shown in Figure 5. There are bad tiles in four rows and avoiding any of the rows of bad tiles takes two steps. For example, compare the following two paths:

1. $(1,0) \rightarrow (1,1) \rightarrow (1,2)$
2. $(1,0) \rightarrow (2,0) \rightarrow (2,1) \rightarrow (2,2) \rightarrow (1,2)$

Path 2 avoids the bad tiles but it takes 4 steps to get to $(1, 2)$ from $(1, 0)$ compared to only 2 steps of Path 1. Since avoiding any tiles costs the same number of extra steps, a natural policy in this maze would be avoiding $HH$ tiles and ignoring $hh$ tiles. However, this is not possible with either

```
                          MAZE
                     ________________
                     |__|G_|__|  10
                     |HH|HH|__|  9
                     |__|__|__|  8
                     |__|hh|hh|  7
                     |__|__|__|  6
                      __|__|__   5
                     |__|__|__|  4
                     |__|hh|hh|  3
                     |__|__|__|  2
                     |HH|HH|__|  1
                     |__|S_|__|  0
                      0   1   2
```

Figure 5: An example MAZE which can be used to demonstrate the issues with uniform thresholding for TLQ.

thresholding method. Now, we will discuss how each thresholding method fails this achieving this Pareto optimal policy.

The action-values show the reward in $G$ discounted by the length of the shortest path to $G$ from the cell this state-action pair leads to. For example, $Q((1,8), Right) = R\gamma^3$ as it takes 3 steps to get to $G$ from $(2,8)$. Hence, the action-values increase as the agent gets closer to the goal. Assume that the agent is in $(1,0)$, the action that we need to take is $Right$, meaning $\tau_1$ should be set smaller than or equal to $\gamma^{11}R$ in Absolute Thresholding. However, since the action values will be larger than this in the states closer to $G$, it will mean that the primary objective will be ignored for the rest of the episode. Hence, the agent will avoid $h$ tiles too and the desired policy is unattainable.

Similarly, since Relative Thresholding effectively limits the length of detours and detours for avoiding $h$s are of the same length as the ones for $H$s, this cannot give a policy that only goes through $h$s.

Absolute Slacking will cause this problem in the reverse, meaning the late episode detours requires detours during the whole episode. Assume the agent is in cell $(1,8)$, then we need

$$Q_1^\star((1,8), R) > Q_1^\star((1,8), U) - \delta_1$$
$$\implies \gamma^3 R > \gamma R - \delta_1$$
$$\implies \delta_1 > R\gamma(1 - \gamma^2)$$

Then, if going left instead of up is not allowed in cell $(1,6)$:

$$Q_1^\star((1,6), R) < Q_1^\star((1,6), U) - \delta_1$$
$$\implies \gamma^5 R < \gamma^3 R - \delta_1$$
$$\implies \delta_1 < R\gamma^3(1 - \gamma^2)$$

Combining these two requirements gives $R\gamma^3(1 - \gamma^2) > R\gamma(1 - \gamma^2)$, which requires $\gamma > 1$ which is false.

## D  VARIATIONS TO TLQ AND SOME ALTERNATIVES

In this section, we will try to address the problems with TLQ within the framework of value function algorithms. We will start by briefly talking about two of our failed attempts (one completely failing and another half-working) to develop working TLQ variants to show the breadth of the problems and our work. Moreover, we believe these ideas are quite natural and can look promising; so, we would like to share our experience to help people working on TLQ algorithms.

Then, we will describe two of our working solutions. While these solutions are limited either in terms of convergence or applicable domains, they provide either a good solution to a sizeable subset of common tasks or give a good alternative to TLQ in the general case.

## D.1 Failed Attempts

In this section, we will give our not very successful attempts to improve the performance of TLQ.

### D.1.1 TL-SARSA

Our first failed attempt was switching to an on-policy learning framework which could solve agents getting stuck problem in Section 4.1. An important reason for this issue was agents' optimistic expectation that they would be following the optimal behavior after each action. So, we considered an on-policy agent which actually uses its realistic behavior to learn could solve our issues.

So, we modified our update functions from Section 4 to mimic SARSA (Sutton & Barto (2018)) instead of Q-Learning. This would mean replacing the max operators with the actual action. For example, the update function from Li & Czarnecki (2019)

$$Q_i^\star(s, a) = \sum_{s' \in S} P(s, a, s')(R_i(s, a, s') + \gamma \max_{\pi \in \Pi_{i-1}} Q_i^\star(s', \pi(s'))) \tag{9}$$

will become:

$$Q_i^\star(s, a) = \sum_{s' \in S} P(s, a, s')(R_i(s, a, s') + \gamma Q_i^\star(s', a')) \tag{10}$$

where $a' = \pi(s')$.

However, this naive attempt failed due to some theoretical limitations of SARSA. Singh et al. (2000) states that the convergence of SARSA is guaranteed under the condition that the policy is greedy in limit. However, our policies are not necessarily greedy with respect to $Q^\star$ in limit. Thresholding means that sometimes actions suboptimal w.r.t. $Q^\star$ are chosen. For example, if $Q_1^\star(s, a_1) > Q_1^\star(s, a_2) > \tau_1$ and $Q_2^\star(s, a_2) > Q_2^\star(s, a_1)$ for a state $s$ in a two objective task, the policy we want to learn is not greedy w.r.t. $Q_1^\star$. This manifested itself as constant oscillations in the policy in our experiments.

### D.1.2 Cyclic Action Selection

Our second half-failed attempt was modifying the action selection mechanism to solve the phenomenon described in Section C.1. It was based on the intuition that the reason for this issue was unnecessary sacrifices in the primary objective that is not required by the secondary objective. For example, if we consider the maze in Figure 1, going left or right in cell $(2, 2)$ is the same w.r.t. secondary objective, hence the agent should not sacrifice from the primary objective irrespective of the thresholds/slacks. Using this intuition, we developed a cyclic action selection algorithm. In Algorithm 3, we show a two objective version of it for simplicity. While it can be generalized to $K$ objectives, we do not believe it would be of interest considering its failure to completely address our problems.

---

**Algorithm 3** CyclicActionSelection

---

**Function** `CyclicActionSelection`$(s, Q|A)$**:**

    $A_0 \leftarrow A$

    $A_1 \leftarrow$ `AcceptableActs`$(s, Q_1, A_0)$

    **if** $|A_1| \leq 1$ **then**         // If there are not more than one option

1     |   **return** $\arg\max_{a \in A_0} Q_1(s, a)$

2     **end**

3     $A_2 \leftarrow$ `AcceptableActs`$(s, Q_2, A_1)$

    **if** $|A_2| \leq 1$ **then**

4     |   **return** $\arg\max_{a \in A_1} Q_2(s, a)$

5     **end**

6     **return** $\arg\max_{a \in A_2} Q_1(s, a)$ ;         // Max over $A_2$ but wrt $Q_1$

---

As the pseudocode shows, the idea is to assign the unconstrained function a threshold/slack which would be used to return the action selection right back to the primary objective but after applying this new threshold. This can be seen from the $Q$-functions used with $\arg\max$ and `AcceptableActs` throughout the algorithm. It starts with using $Q_1$, then uses $Q_2$ if there are multiple acceptable actions w.r.t. $Q_1$. Finally, it uses $Q_1$ again if there is more than one action acceptable w.r.t. $Q_2$. Notice that there is a `AcceptableActs` call using $Q_2$ which is different than Algorithm 1. This requires having a threshold/slacking for the unconstrained objective which is basically a hack. It can be used with any thresholding function from Section B.

However, there are several problems with this approach. Firstly, having a threshold for the unconstrained objective removes one of the important supposed benefits of TLQ, namely its intuitiveness. Especially the cyclic nature of this makes two different threshold values, one for each objective, to be coupled in a complex way when deciding which detours will be taken. This can lead to a blind hyperparameter search. Our experiments show that the success of the policy is highly sensitive to the choices of these two hyperparameters.

Also, the problem described in Section C.2 persists, which means some of the very natural policies cannot be found with this technique.

## D.2    INFORMED TARGETS

While using the SARSA variant has failed as seen in Section D.1.1, we believe that our intuition about the root of the issues described in Section C.1 was correct. Hence, we decided that an approach that could better align the update target with the "actual policy" could still solve the problem with short-sighted sacrifices. One such way could be accounting for the possibility of actions not being taken according to the given objective. Here, we will present the approach for two objectives. Its generalization to $K$ objectives is not necessarily straightforward and we regard it as a future research direction. To illustrate the idea, assume that $Q_1^\star(s', a_1) > Q_1^\star(s', a_2) > \tau_1$ and $Q_1^\star(s', a_2) > Q_2^\star(s', a_1)$ for a state $s'$ in a task with only two actions. Eq. 9 uses $R_1(s, a, s') + \gamma_1(s', a_1)$ when computing update target for $Q_1^\star(s, a)$ as $a_1$ maximizes $Q_1^\star$ in state $s'$. However, this is misleading as $a_1$ will never be chosen in state $s'$. Instead $Q_1^\star(s', a_2)$ should be used as $a_2$ maximizes $Q_2^\star$ in $s'$. Notice that this is still different than TL-SARSA as we may be following a completely different policy. In other words, $a_2$ is used not because it is actually the action taken but it would be the action taken in the optimal case. We call this "informed targets" as value functions make "informed" updates, knowing what would be the actual action taken. More formally, this means modifying the update function for the primary objective to:

$$Q_1^\star(s, a) = \sum_{s' \in S} (R_1(s, a, s') + Q_1^\star(s', \arg\max_{\pi \in \Pi_1} Q_2^\star(s', \pi(s'))) P(s, a, s') \tag{11}$$

Notice that the target for the objective 1 is computed by choosing the optimal action with respect to 2. This prevents optimistic updates that happen due to targets computed with actions that never would be taken. It should be noted that these updates were the reason for the failure mode discussed in Section C.1. Preventing them solves this issue but brings a different problem: Instability in update targets. Consider the scenario in Section C.1. If the current policy is going to left in state $(2, 2)$, the value of going right would be $\gamma R$. Assuming that the threshold is smaller than $\gamma R$, at some point the value of going right would pass the threshold and both going right and left would be equally good. Once this happens, the update target for going right will become $\gamma Q(s, Right)$, hence it will start to decrease until it is smaller than the threshold. Then, the target will go back to its original value, hence resulting in an endless cycle. While it is possible to introduce some buffer in these updates such that the oscillations do not affect the policy that is being followed, the optimality of the resulting policy will depend on the initialization.

The update function with buffer hyperparameter $b$ can be obtained by replacing $\Pi_i$ in Section 11 with $\hat{\Pi}_i$ which is defined as:

$$\hat{\Pi}_i \triangleq \{\pi_i \in \hat{\Pi}_{i-1} \mid Q_i^\star(s, \pi_i(s)) \geq \max_{a \in \{\pi_{i-1}(s) \mid \pi_{i-1} \in \Pi_{i-1}\}} \hat{Q}_i^\star(s, a) - \delta_i - b, \forall s \in \mathcal{S}\} \tag{12}$$

Notice that this will lead to a smaller oscillation zone which in turn is going to prevent the policy from oscillating as it still uses $\Pi_i$. Also, note that the problems in Section C.2 still persists.

### D.3 State Augmentation for Non-reachability Constrained Objective Case

In this section, we will show how a problem where constrained objectives are non-reachability can be solved by augmenting the state space. This idea of state augmentation has been used before with slightly different or narrower purposes Geibel (2006).

#### D.3.1 Single Constrained Objective

When the constrained objective is a non-reachability objective, this can be solved by using state augmentation that keeps track of obtained cost/reward for the constrained objective so far. In this section, we will use a different MDP that is inspired by a real-life scenario to also give a more intuitive example and show the real use of LMDPs.

**An example:** A car travels across the country using highways. It starts the journey in the city $s_0$ and tries to go to a city $s_F \in S_F$. Once he reaches a city in the set $s_F$, he will stop traveling. The highway toll for the highway from the city $s$ to $s'$ is represented by the function $h(s, s')$ where $h : S \times S \to \mathbb{R}$. The driver has a budget of $B$ dollars and tries to have the best trip within this budget. His cost within this budget will be reimbursed by his company, so he has no incentive to spend less as long as he is within the budget. His pleasure from arriving in the city $s$ is given by $p(s)$ where $p : S \to \mathbb{R}$ and $p(s) = 0, \forall s \notin S_F$.

Formally, we have two objectives: minimizing the tolls and maximizing pleasure. Minimizing tolls is constrained/thresholded by the budget $B$. Maximizing the pleasure is unconstrained. Following our formulation in Section 3:

- $R_1(s, a, s') = -h(s, s')$ and $\tau_1 = -B$. Notice again that we expressed the threshold without discounting. Since we will not be using TLQ, we do not need to find the corresponding discounted threshold. Also, notice that the corresponding discounted threshold actually depends on the trajectory.
- $R_2(s, a, s') = p(s')$.

We can express this two-objective task and preserve the preferences by constructing the following single-objective task:

- Set of states: $\hat{S} = S \times \mathbb{R}$ where $(s, c)$ means the driver is in the city $s$ and so far the driver has spent $B - c$ dollars on tolls. Augmented initial state $\hat{s}_0 = (s_0, B)$.
- Set of actions: $\hat{A} = A$
- Transition function $\hat{P} : \hat{S} \times A \times \hat{S} \to [0, 1]$ where

$$\hat{P}((s, c), a, (s', c')) = \begin{cases} P(s, a, s'), c' = c - h(s, s') \\ 0, \text{otherwise} \end{cases} \tag{13}$$

- Reward function $\hat{R} : \hat{S} \times A \times \hat{S} \to \mathbb{R}$

$$\hat{R}((s, c), a, (s', c')) = \begin{cases} 0, \text{if } s' \notin S_F \\ p(s'), \text{else if } c' \geq 0 \\ \lambda c', \text{otherwise} \end{cases} \tag{14}$$

Note that a non-zero reward will be given only when the car reaches a final destination. If the driver has stayed within the budget, he gets his pleasure value as the reward. If he has exceeded the budget, he is penalized accordingly with a multiplier $\lambda$. Implicitly, we assume that $p(s) > 0, \forall s \in S_F$.

Also, note that while this reward function specifies the optimal policy correctly, it may not be a good reward function for learning and exploration purposes. For instance, until the agent learns how to stay within the budget, all the terminal states will have negative values and non-terminal states will have higher values. Hence, the agent can get stuck here by trying to avoid terminal states. Realizing that it can get positive rewards may require a good exploration policy.

This has the following advantages:

- Different thresholds are supported
- Thresholding is intuitive
- Convergence proofs exist.

### D.3.2 Optimality of New MDP

We can easily show that this single-objective task has the same ordering of trajectories as the original task. More formally, $\zeta^1 = s_0^1, a_0^1, s_1^1, a_1^1, \ldots, s_{n^1}^1$ is better than $\zeta^2 = s_0^2, a_0^2, s_1^2, a_1^2, \ldots, s_{n^2}^2$ under the original task if and only if the augmented trajectory $\hat{\zeta}^1$ is also better than augmented trajectory $\hat{\zeta}^2$ under this single-objective task. Here, we will use the cumulative reward as the optimality metric when comparing trajectories. For the original task trajectories, we use the thresholded lexicographic comparison relation defined in Section 3. Note that subscripts $n^1$ and $n^2$ denotes the indexes of $\zeta^1$ and $\zeta^2$, not the polynomials.

**Proof:** $\zeta^1 \geq \zeta^2$ under the original task if and only if one of the following must be true:

1. $\sum_{\zeta^1} R_1(s, a, s'), \sum_{\zeta^2} R_1(s, a, s') \geq \tau_1$ and $\sum_{\zeta^1} R_2(s, a, s') \geq \sum_{\zeta^2} R_2(s, a, s')$

2. $\sum_{\zeta^1} R_1(s, a, s') \geq \tau_1 > \sum_{\zeta^2} R_1(s, a, s')$

3. $\tau_1 > \sum_{\zeta^1} R_1(s, a, s') \geq \sum_{\zeta^2} R_1(s, a, s')$

We can show that each of these statements implies that the same ordering holds for $\hat{\zeta}^1 \geq \hat{\zeta}^2$ under the single objective task. Firstly observe that:

$$\sum_{\hat{\zeta}} \hat{R}(\hat{s}, \hat{a}, \hat{s}') = p(\hat{s}_n(s))$$

$$\iff \sum_{\hat{\zeta}} \hat{R}(\hat{s}, \hat{a}, \hat{s}') > 0$$

$$\iff \hat{s}_n(s) \in S_F \wedge \hat{s}_n(c) \geq 0$$

$$\iff \sum_{\hat{\zeta}} h(s, s') \geq B \iff \sum_{\zeta} R_1(s, a, s') \geq \tau_1$$

Then, for the first case:

$$\sum_{\zeta^1} R_1(s, a, s'), \sum_{\zeta^2} R_1(s, a, s') \geq \tau_1$$

$$\implies \sum_{\hat{\zeta}^1} \hat{R}(\hat{s}, \hat{a}, \hat{s}') = p(\hat{s}_{n^1}(s)) \wedge \sum_{\hat{\zeta}^2} \hat{R}(\hat{s}, \hat{a}, \hat{s}') = p(\hat{s}_{n^2}(s))$$

Also,

$$\sum_{\zeta^1} R_2(s, a, s') \geq \sum_{\zeta^2} R_2(s, a, s')$$

$$\implies p(\hat{s}_{n^1}(s)) \geq p(\hat{s}_{n^1}(s))$$

$$\implies \sum_{\hat{\zeta}^1} \hat{R}(\hat{s}, \hat{a}, \hat{s}') > \sum_{\hat{\zeta}^2} \hat{R}(\hat{s}, \hat{a}, \hat{s}')$$

$$\implies \hat{\zeta}^1 \geq \hat{\zeta}^2$$

For the second case,

$$\sum_{\zeta^1} R_1(s, a, s') \geq \tau_1 > \sum_{\zeta^2} R_1(s, a, s')$$

$$\implies \sum_{\hat{\zeta}^1} \hat{R}(\hat{s}, \hat{a}, \hat{s}') > 0 \wedge \sum_{\hat{\zeta}^2} \hat{R}(\hat{s}, \hat{a}, \hat{s}') \leq 0$$

$$\implies \sum_{\hat{\zeta}^1} \hat{R}(\hat{s}, \hat{a}, \hat{s}') > \sum_{\hat{\zeta}^2} \hat{R}(\hat{s}, \hat{a}, \hat{s}')$$

$$\implies \hat{\zeta}^1 \geq \hat{\zeta}^2$$

For the third case, we can observe that:

$$\tau_1 > \sum_{\zeta^1} R_1(s, a, s') \geq \sum_{\zeta^2} R_1(s, a, s')$$

$$\implies B > \sum_{\hat{\zeta}^1} h(s, s') \geq \sum_{\hat{\zeta}^2} h(s, s')$$

$$\implies \hat{s}_{n^1}(c) \geq \hat{s}_{n^2}(s)$$

$$\implies \lambda \hat{s}_{n^1}(c) \geq \lambda \hat{s}_{n^2}(s)$$

$$\implies \sum_{\hat{\zeta}^1} \hat{R}(\hat{s}, \hat{a}, \hat{s}') > \sum_{\hat{\zeta}^2} \hat{R}(\hat{s}, \hat{a}, \hat{s}')$$

$$\implies \hat{\zeta}^1 \geq \hat{\zeta}^2$$

$\square$

Note that we've specified the unconstrained objective as a quantitative reachability objective, ie. it is non-zero only in the terminal states. Now, we will remove the restriction over $p$.

**Alternative 1:** First option is to extend the state space again to keep track of $p$ as well. So, the MDP will be:

- State Space: $\hat{\hat{S}} = S \times \mathbb{R} \times \mathbb{R}$ where the state $(s, c, \bar{p})$ corresponds to accumulating $\bar{p}\, p(s)$ so far.
- Transition function: $\hat{\hat{P}} : \hat{\hat{S}} \times A \times \hat{\hat{S}} \to [0, 1]$ where

$$\hat{\hat{P}}((s, c, \bar{p}), a, (s', c', \bar{p}')) \tag{15}$$

$$= \begin{cases} \hat{P}((s, c), a, (s', c')), \bar{p}' = \bar{p} + p(s') \\ 0, \text{otherwise} \end{cases} \tag{16}$$

- Reward function: $\hat{\hat{R}} : \hat{\hat{S}} \times A \times \hat{\hat{S}} \to \mathbb{R}$ where

$$\hat{\hat{R}}((s, c, \bar{p}), a, (s', c', \bar{p}')) = \begin{cases} 0, \text{if } s' \notin S_F \\ \bar{p}', \text{else if } c' \geq 0 \\ \lambda c', \text{otherwise} \end{cases} \tag{17}$$

**Alternative 2:** Extending the state space is not always optimal, as it increases the complexity. Instead, we can try to directly modify 14. With this, we will still use $\hat{S}$ and $\hat{P}$ as the state space and transition function, respectively.

- Most simply, we can start giving $p(s')$ reward in the non-terminal states. Then, we can guarantee the lexicographic ordering by subtracting a large value $C_l$ that is guaranteed to be larger than $\sum_t p(s')$ from $\lambda c'$.

$$\hat{\hat{R}}((s, c), a, (s', c')) = \begin{cases} p(s'), \text{if } s' \notin S_F \\ p(s'), \text{else if } c' \geq 0 \\ \lambda c' - C_l, \text{otherwise} \end{cases} \tag{18}$$

Optimality proofs of these new MDPs are very similar to our proof in Section D.3.2. So, we leave it to the reader to avoid repeating it.

### D.3.3 MULTIPLE CONSTRAINED OBJECTIVES CASE

Our analysis above assumes that there are only two objectives: a constrained primary objective and an unconstrained secondary objective. However, in that setting, many CMDP algorithms are readily applicable. Therefore, we are more interested when we have multiple constraints that need to be solved in the lexicographic order. Yet, extending the approach above to this setting is not straightforward. To apply it, we need to know which constraints can be satisfied together. More formally, if the constrained objectives are $1, \ldots, (k-1)$, we need to find the maximum $i$ such that there exists a policy that satisfies objectives $1, \ldots, (i-1)$, i.e. can reach a state $(s, c_1, c_2, \ldots, c_{i-1})$ such that $s \in S_F$ and $c_1, c_2, \ldots, c_{i-1} \geq 0$. We identified three different approaches that could be used for this but we believe future work is needed to develop more efficient methods.

**One-by-one**   The simplest method to solve tasks with multiple constrained objectives is reminiscent of linear search algorithm. We can start with the first (most important) constraint and see if we can find a policy that satisfies it, i.e. can reach $(s, c_1)$ such that $s \in S_F$ and $c_1 \geq 0$ from $s_{init}$. If such a policy exists, we can introduce the second constraint to see if a policy that satisfies both of them simultaneously exists. Continuing in this fashion, it can be found up to which objective the agent can satisfy simultaneously. However, this method can be prohibitively expensive as it requires solving $O(k)$ subproblems. More importantly, it is very hard if not impossible to know whether a subproblem is not solvable or just taking too long to learn.

For this method, we can construct the following reward function for each $i$ value in different ways. An approach would be maximizing the worst violated constraint:

$$\hat{R}((s, c_1, \ldots, c_{i-1}), a, (s', c'_1, \ldots, c'_{i-1})) = \tag{19}$$

$$\begin{cases} R(s, a, s'), \text{if } s' \notin S_F \\ R(s, a, s'), \text{else if } c'_j \geq 0 \ \ \forall j < i \\ \lambda \min_j c'_j - C_l, \text{otherwise} \end{cases} \tag{20}$$

Where $R$ is the reward function of the unconstrained objective in the original MDP and $C_l$ is an upper bound on the unconstrained reward that can be collected during an episode.

**Binary Search**   As the name suggests, this method is inspired by binary search algorithm. Assuming the constrained objectives are $1, \ldots, (k-1)$, we can start by trying to solve constraints $1, \ldots, \lfloor \frac{k}{2} \rfloor$, then we can try $1, \ldots, \lfloor \frac{3k}{4} \rfloor$ or $1, \ldots, \lfloor \frac{k}{4} \rfloor$ depending on whether it was solvable or not, respectively. While this method is faster than one-by-one, it still suffers from the same halting problem. We can use Eq. 19 for this approach too.

**Dynamic Search**   This method is not concretized and is intended mostly as an idea for future research. Hayes et al. (2020) presents an approach to set the threshold values for TLQ dynamically, depending on the attainable performance up to that point in the training. Similarly, we can introduce and remove constraints dynamically during the training without waiting for the algorithm to successfully converge for a subproblem.

## E   CONE PROJECTION

In this section, we show how the projection equation in Eq. 4 is derived. For the sake of completeness, we start with some simpler and well-known projections and move to the derivation of Eq. 4.

### E.1   ORTHOGONAL PROJECTION ONTO A HYPERPLANE

One of the most well-known projection tasks is projecting a vector $y \in \mathbb{R}^n$ onto a hyperplane $H_a$ that passes through origin, specified by its normal vector $a \in \mathbb{R}^n$ as $H_a = \{x \in \mathbb{R}^n \mid \langle x, a \rangle = 0\}$

where $\langle \rangle$ denotes the dot product defined as $\boldsymbol{v}^T \boldsymbol{a} = \sum_i \boldsymbol{v}_i \boldsymbol{a}_i$. Projection of $\boldsymbol{y}$ onto $H_{\boldsymbol{a}}$ is notated as $\boldsymbol{P}_{\boldsymbol{a}}^H(\boldsymbol{y})$ and defined as $\boldsymbol{P}_{\boldsymbol{a}}^H(\boldsymbol{y}) = \arg\min_{\boldsymbol{v} \in H_{\boldsymbol{a}}} \|\boldsymbol{v} - \boldsymbol{P}_{\boldsymbol{a}}^H(\boldsymbol{y})\|$. $\| \|$ denotes the L2 norm defined as

$$\|\boldsymbol{v}\| = \sqrt{\boldsymbol{v}^T \boldsymbol{v}} = \sqrt{\sum_i \boldsymbol{v}_i^2}$$

$\boldsymbol{P}_{\boldsymbol{a}}^H(\boldsymbol{y})$ can be found easily by using well-known result $\boldsymbol{y} - \boldsymbol{P}_{\boldsymbol{a}}^H(\boldsymbol{y}) \parallel \boldsymbol{a}$, i.e. the projection error is parallel to the normal vector of the hyperplane. Then, there is a $c \in \mathbb{R}$ such that $\boldsymbol{y} - \boldsymbol{P}_{\boldsymbol{a}}^H(\boldsymbol{y}) = c\boldsymbol{a}$.

$$\boldsymbol{y} - \boldsymbol{P}_{\boldsymbol{a}}^H(\boldsymbol{y}) \parallel \boldsymbol{a}$$
$$\implies \boldsymbol{P}_{\boldsymbol{a}}^H(\boldsymbol{y}) = \boldsymbol{y} - c\boldsymbol{a}$$
$$\implies \langle \boldsymbol{P}_{\boldsymbol{a}}^H(\boldsymbol{y}), \boldsymbol{a} \rangle = \langle \boldsymbol{y}, \boldsymbol{a} \rangle - c\langle \boldsymbol{a}, \boldsymbol{a} \rangle$$
$$\implies 0 = \langle \boldsymbol{y}, \boldsymbol{a} \rangle - c\|\boldsymbol{a}\|^2$$
$$\implies c = \frac{\langle \boldsymbol{y}, \boldsymbol{a} \rangle}{\|\boldsymbol{a}\|^2}$$
$$\implies \boldsymbol{P}_{\boldsymbol{a}}^H(\boldsymbol{y}) = \boldsymbol{y} - \frac{\langle \boldsymbol{y}, \boldsymbol{a} \rangle}{\|\boldsymbol{a}\|^2} \boldsymbol{a}$$

### E.2 PROJECTION ONTO A HALFSPACE

In many cases, we may want to not project a vector that is already on one side of the hyperplane. For example, if we want to project a vector onto a feasible set, the vector that is already in the feasible set should not be projected. This idea can be formalized by extending the definition above to halfspaces. A positive halfspace $S_{\boldsymbol{a}}^+$ is defined as $S_{\boldsymbol{a}}^+ = \{\boldsymbol{x} \in \mathbb{R}^n \mid \langle \boldsymbol{x}, \boldsymbol{a} \rangle \geq 0\}$. This can be thought of as the set of vectors with which $\boldsymbol{a}$ makes an angle less than or equal to $\frac{\pi}{2}$. We can define the projection $\boldsymbol{y}$ onto $S_{\boldsymbol{a}}^+$ as follows:

$$\boldsymbol{P}_{\boldsymbol{y}}^{S^+}(\boldsymbol{a}) = \begin{cases} \boldsymbol{y}, & \boldsymbol{y} \in S_{\boldsymbol{a}}^+ \\ \boldsymbol{P}_{\boldsymbol{y}}^H(\boldsymbol{a}), & \text{otherwise} \end{cases} \tag{21}$$

Note that the piecewise function handles $\boldsymbol{y} \in S_{\boldsymbol{a}}^+$ and $\boldsymbol{y} \notin S_{\boldsymbol{a}}^+$ cases separately.

### E.3 PROJECTING A VECTOR ONTO A CONE

While halfspaces are one of the most common sets in practice, they can be limiting in many cases. A natural extension to this idea would be limiting the set to vectors with which $\boldsymbol{a}$ makes an angle $\frac{\pi}{2} - \Delta$ for some $0 \leq \Delta \leq \frac{\pi}{2}$. The angle between two vectors is defined using dot product:

$$\langle v, u \rangle = \cos \angle v, u \|\boldsymbol{a}\| \|\boldsymbol{x}\|$$

Note that since $\cos(\Delta) = \cos 2\pi - \Delta$, the $\angle v, u$ can take two values between $0$ and $2\pi$. For simplicity, we will always talk about the smaller angle, i.e. $\angle : \mathbb{R}^n \times \mathbb{R}^n \to [0, pi]$.

This would be a hypercone which simplifies to a halfspace when $\Delta = 0$.

Let $C_a^\Delta$ be a hypercone with axis $a \in \mathbb{R}^n$ and angle $\frac{\pi}{2} - \Delta$, i.e.

$$C_a^\Delta = \{\boldsymbol{x} \in \mathbb{R}^n \mid \|\boldsymbol{x}\| = 0 \vee \frac{\boldsymbol{a}^T \boldsymbol{x}}{\|\boldsymbol{a}\| \|\boldsymbol{x}\|} \geq \cos\left(\frac{\pi}{2} - \Delta\right)\} \tag{22}$$

which uses the dot product formula above to see if cosine of the angle between $\boldsymbol{a}$ and $\boldsymbol{x}$ is greater than cosine of $\frac{\pi}{2} - \Delta$. For $0 \leq \Delta \leq \frac{\pi}{2}$, this corresponds to the angle between $\boldsymbol{a}$ and $\boldsymbol{x}$ being in the interval $[0, \frac{\pi}{2} - \Delta]$.

Then, the projection of a vector $\boldsymbol{g} \in \mathbb{R}^n$ onto $C$ is defined as

$$\boldsymbol{g}_C^p = \arg\min_{\hat{\boldsymbol{g}} \in C} \|\hat{\boldsymbol{g}} - \boldsymbol{g}\|_2 \tag{23}$$

Solving this equation is not as straightforward as for the halfspaces. We will first show that $g^p$ is planar with $g$ and $a$, i.e. they can be written as linear combinations of each other. are all on the same plane. This is intuitive and well-known in lower dimensions, but below can be seen a formal proof for higher dimensions. Once this is proven, we can utilize some two-dimensional geometric intuition to simplify the algebra.

### E.4   PROOF OF PLANARITY

The projection is a constrained optimization problem:

$$
\min \|\boldsymbol{x} - \boldsymbol{g}\|_2
$$
$$
\text{subject to} \quad \frac{\boldsymbol{a}^T \boldsymbol{x}}{\|\boldsymbol{a}\|\|\boldsymbol{x}\|} \geq \cos\left(\frac{\pi}{2} - \Delta\right)
$$

If we can show that the solution to this vector is planar with $g$ and $a$, we will be done. The solution to this constrained optimization problem should satisfy Karush-Kuhn-Tucker (KKT) conditions which generalize the Lagrange Multiplier method to problems with inequality constraints. However, applying KKT conditions in this format does not provide a clean result. Therefore, we will prove a stronger claim that gives cleaner KKT conditions:

**Lemma E.1.** *For any fixed length $\boldsymbol{x}$, the projection is minimized when $\boldsymbol{x}$, $\boldsymbol{g}$, and $\boldsymbol{a}$ are planar.*

*Proof.* This gives us the following modified optimization problem with an additional constraint. Now, we will show that the planarity does not depend on $\|\boldsymbol{x}\|$, which will be denoted as $c$.

$$
\min \quad f(\boldsymbol{x}) = \|\boldsymbol{x} - \boldsymbol{g}\|_2
$$
$$
\text{subject to} \quad r(\boldsymbol{x}) = \frac{\boldsymbol{a}^T \boldsymbol{x}}{\|\boldsymbol{a}\|\|\boldsymbol{x}\|} \geq \cos\left(\frac{\pi}{2} - \Delta\right) = \sin\Delta
$$
$$
h(\boldsymbol{x}) = \|\boldsymbol{x}\| = c
$$

Swapping norms with their dot product equivalents (replacing the norm in the objective and equality constraint with a norm square for conciseness) and writing the remaining in the standard format gives us:

$$
\min \quad f(\boldsymbol{x}) = \boldsymbol{x}^T \boldsymbol{x} - 2\boldsymbol{g}^T \boldsymbol{x} + \boldsymbol{g}^T \boldsymbol{g}
$$
$$
\text{subject to} \quad r(\boldsymbol{x}) = \sin\Delta - \frac{\boldsymbol{a}^T \boldsymbol{x}}{\sqrt{\boldsymbol{a}^T \boldsymbol{a}}\sqrt{\boldsymbol{x}^T \boldsymbol{x}}} \leq 0
$$
$$
h(\boldsymbol{x}) = \boldsymbol{x}^T \boldsymbol{x} - c^2 = 0
$$

KKT conditions for this problem require that any minimum point $\hat{\boldsymbol{x}}$ should satisfy the following condition Chong & Zak (2004):

$$\nabla f(\hat{\boldsymbol{x}}) + \lambda \nabla h(\hat{\boldsymbol{x}}) + \mu \nabla r(\hat{\boldsymbol{x}}) = \boldsymbol{0}$$

$$\implies 2\hat{\boldsymbol{x}} - 2\boldsymbol{g} + \lambda 2\hat{\boldsymbol{x}} + \mu \frac{\boldsymbol{a}(\sqrt{\boldsymbol{a}^T\boldsymbol{a}}\sqrt{\hat{\boldsymbol{x}}^T\hat{\boldsymbol{x}}}) - \frac{1}{2}\frac{\sqrt{\boldsymbol{a}^T\boldsymbol{a}}}{\sqrt{\hat{\boldsymbol{x}}^T\hat{\boldsymbol{x}}}}2\hat{\boldsymbol{x}}(\boldsymbol{a}^T\hat{\boldsymbol{x}})}{(\hat{\boldsymbol{x}}^T\hat{\boldsymbol{x}})(\boldsymbol{a}^T\boldsymbol{a})} = \boldsymbol{0} \qquad \hat{\boldsymbol{x}}^T\hat{\boldsymbol{x}} = c^2 \quad \text{from feasibility}$$

$$\implies 2\hat{\boldsymbol{x}} - 2\boldsymbol{g} + \lambda 2\hat{\boldsymbol{x}} + \mu \frac{\boldsymbol{a}(c\sqrt{\boldsymbol{a}^T\boldsymbol{a}}) - \frac{\sqrt{\boldsymbol{a}^T\boldsymbol{a}}}{c}\hat{\boldsymbol{x}}(\boldsymbol{a}^T\hat{\boldsymbol{x}})}{(c^2)(\boldsymbol{a}^T\boldsymbol{a})} = \boldsymbol{0}$$

$$\implies 2\hat{\boldsymbol{x}} - 2\boldsymbol{g} + \lambda 2\hat{\boldsymbol{x}} + \mu \frac{c\boldsymbol{a} - \frac{\boldsymbol{a}^T\hat{\boldsymbol{x}}}{c}\hat{\boldsymbol{x}}}{c^2\sqrt{\boldsymbol{a}^T\boldsymbol{a}}} = \boldsymbol{0} \qquad\qquad \frac{\boldsymbol{a}^T\hat{\boldsymbol{x}}}{\sqrt{\hat{\boldsymbol{x}}^T\hat{\boldsymbol{x}}}} = \sin\Delta\sqrt{\boldsymbol{a}^T\boldsymbol{a}} \quad \text{C.S.}$$

$$\implies 2\hat{\boldsymbol{x}} - 2\boldsymbol{g} + \lambda 2\hat{\boldsymbol{x}} + \mu \frac{c\boldsymbol{a} - \sin\Delta\sqrt{\boldsymbol{a}^T\boldsymbol{a}}\hat{\boldsymbol{x}}}{c^2\sqrt{\boldsymbol{a}^T\boldsymbol{a}}} = \boldsymbol{0}$$

$$\implies 2\hat{\boldsymbol{x}} - 2\boldsymbol{g} + \lambda 2\hat{\boldsymbol{x}} + \mu \frac{\boldsymbol{a}}{c\sqrt{\boldsymbol{a}^T\boldsymbol{a}}} - \mu \frac{\sin\Delta}{c^2}\hat{\boldsymbol{x}} = \boldsymbol{0} \qquad\qquad \text{Reorganize the terms}$$

$$\implies \hat{\boldsymbol{x}}(2 + 2\lambda - \mu\frac{\sin\Delta}{c^2}) = 2\boldsymbol{g} - \mu\frac{\boldsymbol{a}}{c\sqrt{\boldsymbol{a}^T\boldsymbol{a}}}$$

Since $\boldsymbol{g}^p$ is such a minimum point, the above analysis holds for it too. Hence, it can be written as a linear combination of $\boldsymbol{a}$ and $\boldsymbol{g}$. This means that the three vectors are planar.

$\square$

Note that we can see another important result from the analysis above. The complementary slackness condition of KKT requires that $\mu r(\hat{\boldsymbol{x}}) = 0$. However, if $\mu = 0$, the last line equation in the proof simplifies to

$$\hat{\boldsymbol{x}}(2 + 2\lambda) = 2\boldsymbol{g}$$

If $(2 + 2\lambda) \geq 0$, it means $\boldsymbol{g}$ and $\hat{\boldsymbol{x}}$ in the same direction. This is only possible if $\boldsymbol{g}$ is already in the hypercone. If $(2 + 2\lambda) < 0$, this means $\boldsymbol{g}$ and $\hat{\boldsymbol{x}}$ are in the opposite directions which cannot be the projection, as choosing 0 vector would give a smaller projection error. Hence, unless $\boldsymbol{g}$ is already in the hypercone and does not require a projection, $r(\hat{\boldsymbol{x}})$ should be 0. That means the angle between $\boldsymbol{a}$ and $\hat{\boldsymbol{x}}$ is $\frac{\pi}{2} - \Delta$.

### E.5 DERIVATION OF THE PROJECTION FORMULA

Now that it is known that all three vectors are planar, we can just use two-dimensional geometry to reason about it and derive the formula. This can be done as these three vectors in $\mathbb{R}^n$ will span a two-dimensional subspace of $\mathbb{R}^n$ unless they are all collinear, i.e. scalar multiplicative of each other. This would mean that $\boldsymbol{a}$ and $\boldsymbol{g}$ are already in the same direction and no projection is needed, which is a special case we will consider separately. Also, any 2-dimensional subspace of $\mathbb{R}^n$ is isomorphic to $\mathbb{R}^2$, i.e. identical in structurelin (2021). Figure 6 shows the case when the angle between $\boldsymbol{a}$ and $\boldsymbol{g}$, $\phi$, is larger than $\frac{\pi}{2}$. It can be confirmed that the other configurations like will result in the same equations too. Note that when writing the equations below, we considered when $\boldsymbol{g}$ is outside of the cone. When $\boldsymbol{g} \in C$, we will simply call $\boldsymbol{g}^p = \boldsymbol{g}$ similar to the piecewise function in Section E.2.

Firstly, we will find the direction of the projection. Let $\boldsymbol{p}'$ be a vector with the same direction as $\boldsymbol{g}^p$ and it can be written as below.

$$\boldsymbol{p}' = \boldsymbol{g} + \alpha\boldsymbol{a}$$

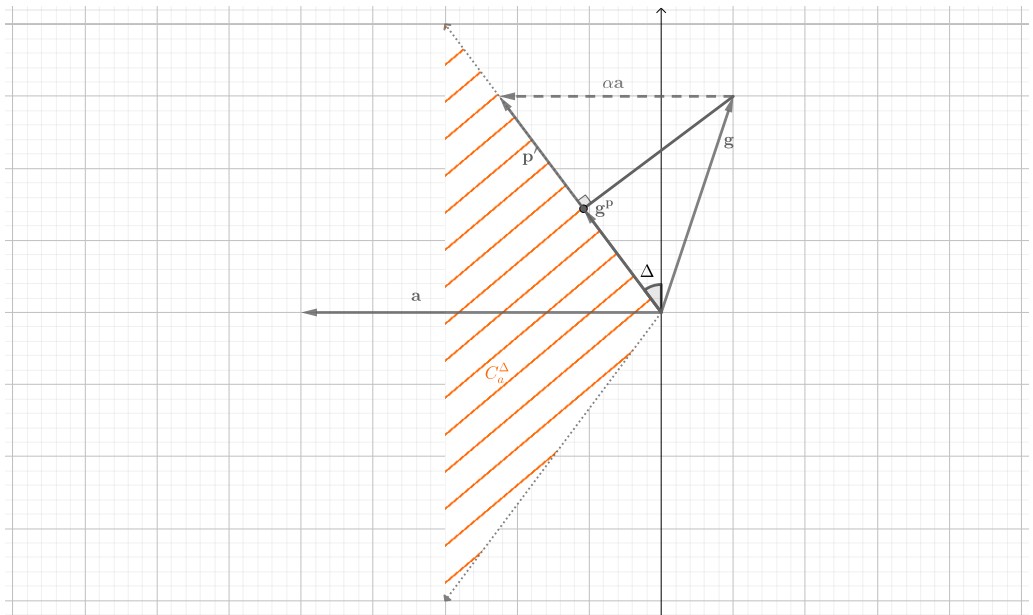

Figure 6: This figure shows how the vectors would be positioned on a plane. The orange region shows the cone. The angle between $\boldsymbol{a}$ and $\boldsymbol{g}$, $\phi$, is omitted to not crowd the figure.

Then, we can find $\alpha$ as below by using the law of sines:

$$\alpha\|\boldsymbol{a}\| = \|\boldsymbol{g}\|\sin\left(\phi - \frac{\pi}{2}\right) + \|\boldsymbol{g}\|\cos\left(\phi - \frac{\pi}{2}\right)\frac{1}{\sin\left(\frac{\pi}{2} - \Delta\right)}\sin\Delta$$

$$\implies \alpha\|\boldsymbol{a}\| = -\|\boldsymbol{g}\|\sin\left(\frac{\pi}{2} - \phi\right) + \|\boldsymbol{g}\|\cos\left(\frac{\pi}{2} - \phi\right)\frac{1}{\sin\left(\frac{\pi}{2} - \Delta\right)}\sin\Delta$$

$$\implies \alpha\|\boldsymbol{a}\| = -\|\boldsymbol{g}\|\cos\phi + \|\boldsymbol{g}\|\sin\phi\frac{1}{\cos\Delta}\sin\Delta$$

$$\implies \alpha\|\boldsymbol{a}\| = \|\boldsymbol{g}\|(\sin\phi\frac{1}{\cos\Delta}\sin\Delta - \cos\phi)$$

$$\implies \alpha = \frac{\|\boldsymbol{g}\|}{\|\boldsymbol{a}\|}(\sin\phi\frac{1}{\cos\Delta}\sin\Delta - \cos\phi)$$

$$\implies \alpha = \frac{\|\boldsymbol{g}\|}{\|\boldsymbol{a}\|}(\sin\phi\tan\Delta - \cos\phi)$$

$$\implies \boldsymbol{p'} = \boldsymbol{g} + \frac{\|\boldsymbol{g}\|}{\|\boldsymbol{a}\|}(\sin\phi\tan\Delta - \cos\phi)\boldsymbol{a}$$

This $\boldsymbol{p'}$ has the correct direction but not necessarily the correct norm to minimize the projection error. The correct projection will be $\boldsymbol{g}^p = k\boldsymbol{p'}$ where $k \in \mathbb{R}$. We can find the $k$ using the well-known rule

that the projection error is perpendicular to the projection.

$$\langle \boldsymbol{g} - k\boldsymbol{p'}, \boldsymbol{p'} \rangle = 0$$
$$\implies \langle \boldsymbol{g}, \boldsymbol{p'} \rangle - k\langle \boldsymbol{p'}, \boldsymbol{p'} \rangle = 0$$
$$\implies \|\boldsymbol{g}\|\|\boldsymbol{p'}\| \cos\left(\Delta + \phi - \frac{\pi}{2}\right) - k\|\boldsymbol{p'}\|^2 = 0$$
$$\implies \|\boldsymbol{g}\|\|\boldsymbol{p'}\| \cos\left(\frac{\pi}{2} - \Delta + \phi\right) - k\|\boldsymbol{p'}\|^2 = 0$$
$$\implies \|\boldsymbol{g}\|\|\boldsymbol{p'}\| \sin\left(\Delta + \phi\right) - k\|\boldsymbol{p'}\|^2 = 0$$
$$\implies \|\boldsymbol{p'}\|(\|\boldsymbol{g}\| \sin\left(\Delta + \phi\right) - k\|\boldsymbol{p'}\|) = 0$$
$$\implies \|\boldsymbol{g}\| \sin\left(\Delta + \phi\right) - k\|\boldsymbol{p'}\| = 0 \qquad \text{if } \|\boldsymbol{p'}\| \neq 0$$
$$\implies \|\boldsymbol{g}\| \sin\left(\Delta + \phi\right) = k\|\boldsymbol{p'}\|$$
$$\implies k = \frac{\|\boldsymbol{g}\|}{\|\boldsymbol{p'}\|} \sin\left(\Delta + \phi\right)$$

The same result also could be obtained by solving another optimization problem with $k$ as the variable. Combining the formula for $\boldsymbol{p'}$ and $k$ gives the formula for $\boldsymbol{g}^p$.sec:supp:pg:exp:reachability

Moving forward, we'll assume a function $projectCone(\boldsymbol{g}, \boldsymbol{a}, \Delta)$ which returns the projection of $\boldsymbol{g}$ onto $C_{\boldsymbol{a}}^{\Delta}$ possibly handling $\boldsymbol{g} \in C_{\boldsymbol{a}}^{\Delta}$ and $\boldsymbol{g} \notin C_{\boldsymbol{a}}^{\Delta}$ cases separately.

# F  LEXICOGRAPHIC PROJECTION ALGORITHM

In this section, we start by giving some background on gradients and directional derivatives that is necessary to understand our algorithm. Then, we share the formulation of the lexicographic optimization problems we solve. Finally, we give some further justification on why we need cone projection instead of halfspace projection and share the remaining results with our algorithm that was left out of the main paper due to space constraints.

## F.1  BACKGROUND ON GRADIENT AND DIRECTIONAL DERIVATIVES

Gradient of a function gives the direction and rate of the fastest increase from point $p$. Moreover, directional derivative of $F$ at $p$ along direction $\boldsymbol{u}$, i.e. $\frac{\partial F}{\partial \boldsymbol{u}}(p)$, can be computed as $\langle \boldsymbol{u}, \nabla F(p) \rangle$.

Intuitively, the directional derivatives give the rate of change $\nabla F(p)$ in the given direction. As can the dot product implies, this rate is the largest when the angle between $\boldsymbol{u}$ and $\nabla F$ is zero. In other words, the gradient gives the direction of the fastest increase.

Using directional derivatives, we can reason about how changes to $p$ affect the value of $F$. For example, since the gradient has the fastest instantaneous rate of change, $F(p + \epsilon \frac{\nabla F(p)}{\|\nabla F(p)\|}) \geq F(p + \epsilon \frac{\boldsymbol{u}}{\|\boldsymbol{u}\|}), \forall \boldsymbol{u} \in \mathbb{R}^n$ for sufficiently small $\epsilon$.

Similarly, if $\angle \boldsymbol{u}, \nabla F(p) \leq \frac{\pi}{2}$, $F(p + \epsilon \boldsymbol{u}) \geq F(p)$ for sufficiently small $\epsilon$. This can be confirmed by computing the directional derivative using $\langle \boldsymbol{u}, \nabla F(p) \rangle = \|u\|\|\nabla F(p)\| \cos \angle \boldsymbol{u}, \nabla F(p)$. In other words, using the directional derivatives, we can obtain a direction of non-decrease for a sufficiently small step size.

## F.2  FORMULATION OF THRESHOLDED LEXICOGRAPHIC MULTI-OBJECTIVE OPTIMIZATION PROBLEMS

A generic multi-objective optimization problem with $K$ objectives and $n$ parameters can be formulated as:

Given a function $F : A \to \mathbb{R}^K$ where $A \in \mathbb{R}^n$ and a comparison relation $\geq^c$ for value vectors in $\mathbb{R}^K$, find an element $\theta^* \in A$ such that $f(\theta^*) \geq^c f(\theta)$ for all $\theta \in A$.

Notice that when we have multiple objectives, the gradients will form a $K$-tuple, $G = (\nabla F_1, \nabla F_2, \cdots, \nabla F_K)$, where $\nabla F_i$ is the gradient of $i^{th}$ component of $F$.

Different instantiations of the comparison relation lead to various multi-objective problem families. In the case of Lexicographic Multi-Objective Optimization, the comparison relation $>^c$, is defined as

$$\boldsymbol{v_1} >^c \boldsymbol{v_2} \iff \exists i < K \text{ s. t. } \forall j < i \; \boldsymbol{v_1}(j) \geq \boldsymbol{v_2}(j))$$
$$\wedge \boldsymbol{v_1}(i) > \boldsymbol{v_2}(i))$$

In *Thresholded* Lexicographic Multi-Objective Optimization, a threshold vector $\tau \in \mathbb{R}^{K-1}$ is introduced to express the values after which the user does not care about improvements in that objective. This new comparison relation can be denoted by $>^{(c,\tau)}$ which is defined as:

$\mathbf{u} >^{\boldsymbol{\tau}} \mathbf{v}$ iff there exists $i \leq K$ such that:

- $\forall j < i$ we have $\mathbf{u_j} \geq \min(\mathbf{v_j}, \tau_j)$; and
    - if $i < K$ then $\min(\mathbf{u_i}, \tau_i) > \min(\mathbf{v_i}, \tau_i)$,
    - otherwise if $i = K$ then $\mathbf{u_i} > \mathbf{v_i}$.

The relation $\geq^{\boldsymbol{\tau}}$ is defined as $>^{\boldsymbol{\tau}} \vee =$.

Notice that this completely parallels the definition of LMDPs from Section 3.

### F.3 JUSTIFICATION OF CONE PROJECTION

Since thresholded lexicographic multi-objective optimization problems impose a strict importance order on the objectives and it is not known how many objectives can be satisfied simultaneously beforehand, a natural approach is to optimize the objectives one-by-one until they reach the threshold values. However, once an objective is satisfied, optimizing the next objective could have a detrimental effect on the satisfied objective. This could even lead to the previous objective failing. While we can always go back to optimizing this failing objective, this would be inefficient, even worse, potentially leading to endless loops of switching between objectives.

However, we could limit our search for a satisfying point for the new objective to the directions not detrimental to already satisfied objectives by using our results about directional derivatives. For simplicity, assume that we have a primary objective $F_1$ which is satisfied at the current point $\theta_n$ and a secondary objective $F_2$ which we are trying to optimize next. $\nabla_{\boldsymbol{u}} F_1$, the change in $F_1$ along a direction $\boldsymbol{u}$, is equal to $\langle \boldsymbol{u}, \nabla F_1(\theta_n) \rangle = \|\boldsymbol{u}\| \|\nabla F_1(\theta_n)\| \cos(\angle \boldsymbol{u}, \nabla F_1(\theta_n))$, choosing a direction which makes an angle $\phi \in [-\frac{\pi}{2}, \frac{\pi}{2}]$ with $\nabla F_1(\theta_n)$ would make the directional derivative non-negative. Therefore, updating $\theta$ as $\theta_{n+1} = \theta_n + \epsilon \boldsymbol{u}$ with an infinitesimal $\epsilon$ would not reduce the value of $F_1$. If $\nabla_{\boldsymbol{u}} F_2$ is positive, we can optimize $F_2$ without jeopardizing $F_1$. Note that the same logic hold even if we have $k$ already satisfied objectives $F_1, \ldots, F_k$ and now optimizing $F_{k+1}$ as long as $\forall i \leq k \nabla_{\boldsymbol{u}} F_i \geq 0$.

While any such $\boldsymbol{u}$ allows us to carefully optimize our new objective $F_2$, we should pick an $\boldsymbol{u}$ with maximum $\frac{\partial F_2}{\partial \boldsymbol{u}}(\theta_n)$ to optimize $F_2$ most efficiently. While we know that $\nabla F_2(\theta_n)$ has the maximum directional derivative, it may not satisfy our previous requirements. Instead, we can use the vector projection to find the $\boldsymbol{u}$ which minimizes $\|\boldsymbol{u} - \nabla F_2(\theta_n)\|$ under the constraint $\nabla_{\boldsymbol{u}} F_1 \geq 0$. Notice that non-negative directional derivative means that $\boldsymbol{u}$ lies on the positive halfspace of $\nabla F_1(\theta_n)$, i.e. $\boldsymbol{u} \in S_{\boldsymbol{u}}^+$. So, projecting $\nabla F_2(\theta_n)$ onto $S_{\nabla F_1(\theta_n)}^+$ will give us the $\boldsymbol{u}$ which satisfies the requirement and is closest to $\nabla F_2 \theta_n$, i.e. has the largest directional derivative. As a special case, when $\nabla F_1$ and $\nabla F_2$ point in opposite directions, this projection will give a zero vector which means that we cannot optimize $F_2$ without sacrificing $F_1$. This point would be locally Pareto optimal. In general, iteratively projecting $\nabla F_{k+1}(\theta_n)$ on the positive halfspaces of $\nabla F_1(\theta_n), \ldots, \nabla F_k(\theta_n)$ gives the desired vector as long as the final vector satisfies the requirements. If it does not satisfy the requirements, this point can be called a locally Pareto optimal point.

While the approach above has the theoretical guarantees for the infinitely small step size, this does not translate to practice as the step sizes are not small enough. For example, Figure 7 shows how a direction that lies on the positive halfspace of the gradient can lead to a decrease. It can be also seen that unless the step size is infinitely small, this would always lead to a decrease. We can overcome this issue by generalizing halfspace to a hypercone for which the central angle is $\frac{\pi}{2} - \Delta$ where $\Delta$

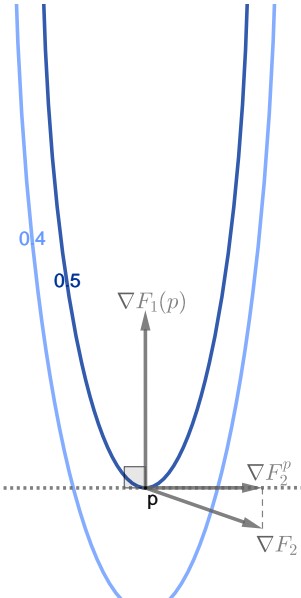

Figure 7: This figure shows why projecting onto the positive hyperspace is not always enough. The curves show the level curves of a function $F_1$. $\nabla F_2^p$ shows the projection of $\nabla F_2$ on the positive halfspace of $\nabla F_1$. Note that following $\nabla F_2^p$ reduces the function from 0.5 to 0.4.

is the hyperparameter of conservativeness. For $\Delta = 0$, this would be the halfspace case introduced above. Figure 8 shows how hypercone projection differs from halfspace projection and keeps the function above or at the current level for reasonably large step sizes.

## F.4 EXPERIMENTS

In this section, we present the rest of the results for the experiments shown in Figure 2. All of the experiments are done using the same benchmark problem described in the main paper.

Figure 9 demonstrates how the value changes shown in Figure 2 were reflected on the parameter space. The figure shows the trajectory the algorithm takes over the level curves of the functions. Notice that $F_2$, in blue, is completely ignored until the threshold for $F_1$ is reached. Then, the algorithm optimizes $F_2$ while respecting the passed threshold of $F_1$, indicated by its trajectory almost along the level curve of $F_1$.

Repeating the same experiment with $AC$ heuristic and $b = 0.01$ yields the results shown in Figure 10 and Figure 11. Notice that the highest value we were able to obtain for $F_2$ was $-0.580$ without $AC$ heuristic; but this was improved to $-0.554$ with $AC$. This is because $AC$ prevents unnecessarily improving $F_1$ over the threshold. This can be observed from the final values of $F_1$ which is $-0.450$ without $AC$ and $-0.496$ with $AC$. The downside is losing the smooth and safe trajectory allowed by our vanilla algorithm, indicated by the zig-zags in Figure 10 and Figure 11. The zig-zags represent the corrections for sacrificing too much from $F_1$ when optimizing $F_2$.

## G USING LEXICOGRAPHIC PROJECTION ALGORITHM IN RL

In this section, we first give the REINFORCE algorithm we use as the basis for our Lexicographic REINFORCE algorithm for easier comparison. Then, we share further details of our experiments.

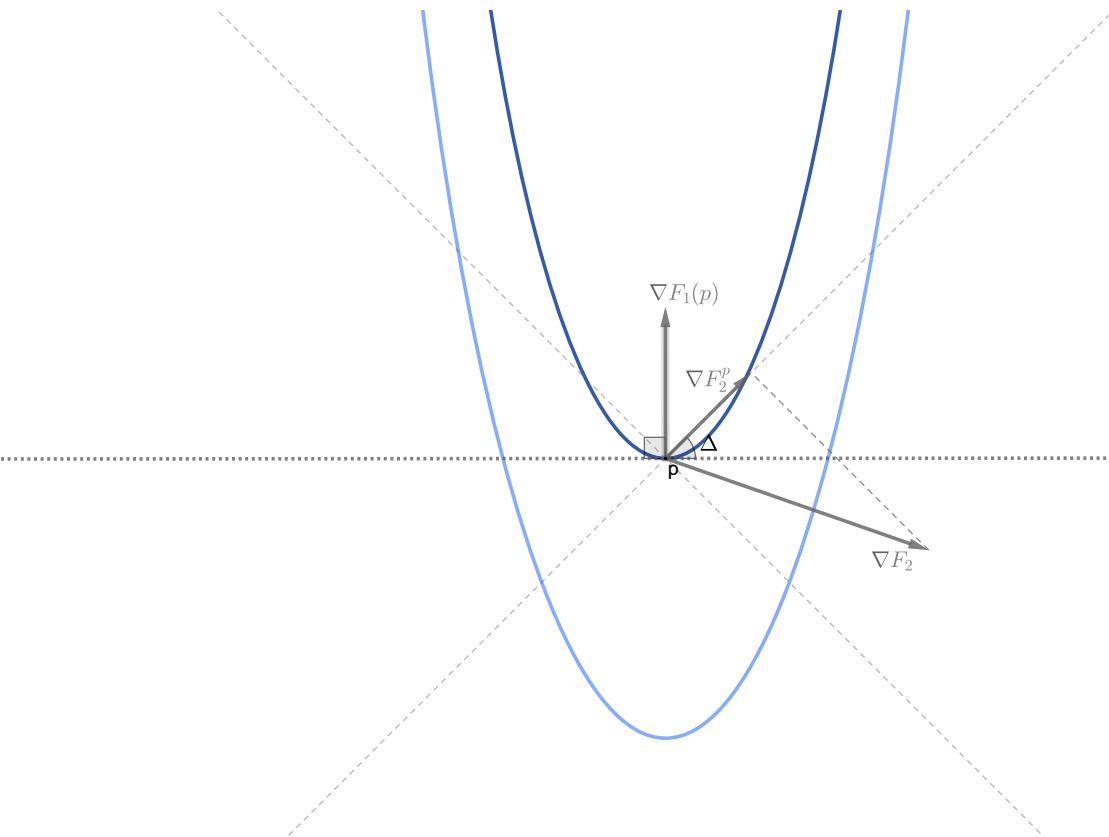

Figure 8: A visualization of cone projection. The dashed line shows the boundaries of the cone which are two lines in two dimensions. Notice that following $\nabla F_2^p$ keeps the function from at or above $0.5$ unless a very large step size is chosen.

### G.1  REINFORCE ALGORITHM

The pseudocode for REINFORCE algorithm that we will use as the basis for our adaptation (Algorithm 5) can be seen in Algorithm 4.

---

**Algorithm 4** Vanilla REINFORCE

---

**Process** `REINFORCE`**:**

    Initialize policy function $\pi(a|s, \theta)$ with random parameter $\theta$

    **for** $ep = 1, N_e$ **do**

        Generate an episode $S_0, A_0, R_1, \ldots, S_{T-1}, A_{T-1}, R_T$ and save $\ln \pi(A_t|S_t)$ at every step.

        $G_{T+1} \leftarrow 0$

        **for** $t = T, 1$ **do**

            $G_t \leftarrow R_t + \gamma G_{t+1}$

        **end**

        $L \leftarrow -\sum_{t=0,T-1} \ln \pi(A_t|S_t)G_{t+1}$

        Update $\theta$ by taking an optimizer step for loss $L$

    **end**

    **return** $\pi(a|s, \theta)$

---

Note that Algorithm 4 can be used with optimizers other than vanilla gradient descent. In our experiments, we found that Adam is easier to use with the tasks at hand. Similarly, we found that using Adam optimizer is better than vanilla gradient descent for our adaptation too.

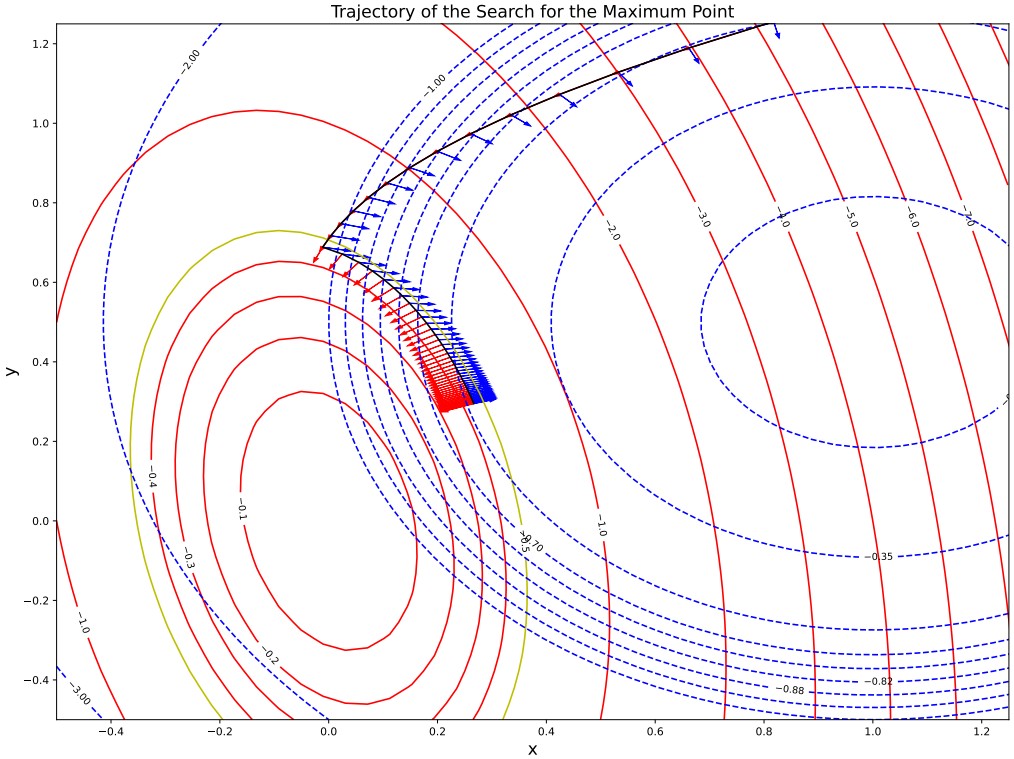

Figure 9: Behavior of the algorithm without the active constraints heuristic and hyperparameters $\alpha = 0.2$ and $\Delta = \frac{\pi}{90}$. The red and blue curves show the level curves of $F_1$ and $F_2$, respectively. The single yellow curve shows the threshold for $F_1$. The black line shows the trajectory of the solution, while the red and blue arrows show the gradients w.r.t. $F_1$ and $F_2$, respectively.

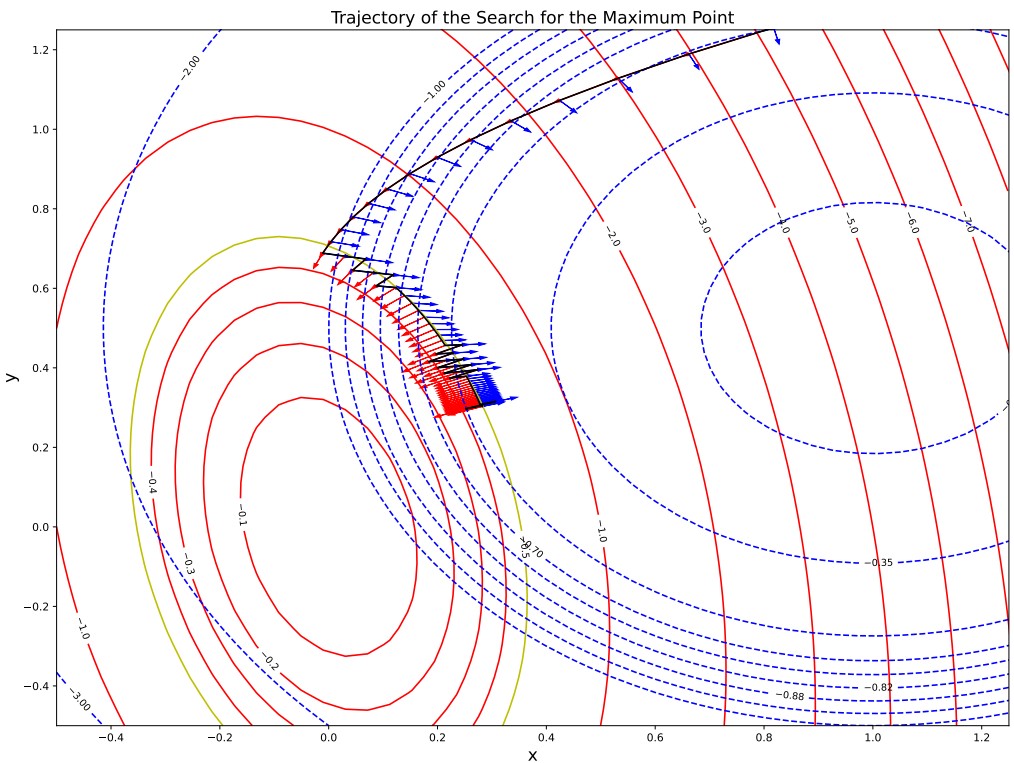

Figure 10: Behavior of the algorithm with the active constraints heuristic and $b = 0.01$. The rest of the hyperparameters are as described in Figure 9.

### G.2 OUR ADAPTATION OF REINFORCE

---
**Algorithm 5** Lexicographic REINFORCE
---
**Process** REINFORCE $(\tau, \Delta, AC, b, N_e)$ **:**
    Initialize policy function $\pi(a|s, \theta)$ with random parameter $\theta$
    **for** $ep = 1, N_e$ **do**
        Generate an episode $S_0, A_0, R_1, \ldots, S_{T-1}, A_{T-1}, R_T$ and save $\ln \pi(A_t|S_t)$ at every step.
        $M \leftarrow \emptyset$
        $F \leftarrow 0$
        **for** $o = 1, K$ **do**
            $G_{T+1} \leftarrow 0$
            **for** $t = T, 1$ **do**
                $G_t \leftarrow R_t + \gamma G_{t+1}$
                $F_o \leftarrow F_o + R_t$
            **end**
            $L \leftarrow -\sum_{t=0, T-1} \ln \pi(A_t|S_t) G_{t+1}$
            Compute the gradient of $L$ with respect to $\theta$ and append it to $M$
        **end**
        $d = $ FindDirection $(M, F, \tau, \Delta, AC, b)$
        Use $d$ as the gradient for the optimizer step to update $\theta$.
    **end**
    **return** $\pi(a|s, \theta)$

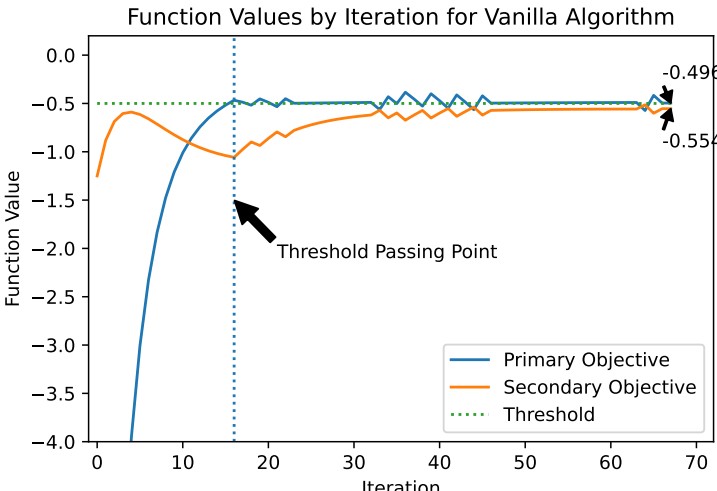

Figure 11: The changes in the function values for the experiment described in Figure 10.

```
            MAZE
        ___________
        |___|G_|___|  4
        |___|hh|hh|  3
        |___|___|___|  2
        |HH|HH|___|  1
        |S_|___|___|  0
         0   1   2
```

Figure 12: The maze to be used in Reachability experiment.

### G.3 EXPERIMENTS

In this section, we share the details of the experimental setup for the adapted REINFORCE algorithm.

#### G.3.1 POLICY FUNCTION

In both experiments, we use a two layer neural network (LeCun et al. (2015)) for policy function. We represent the state via one-hot encoding (Harris & Harris (2015)), hence the input dimension is the same as the size of state space. For example, 20 for the maze in Figure 3. Then the hidden layer is a fully connected layer with 128 units and they use *ReLU* activation function Agarap (2018). We also used a dropout layer (Srivastava et al. (2014)) with drop probability 0.6. Finally, the output layer has 4 units, representing the four valid actions in our benchmark. The outputs of these units are converted to action probabilities by applying a softmax function with temperature 10 LeCun et al. (2015). The temperature hyperparameter allows making the policy less deterministic by making the action probabilities closer to each other. This makes sure that the policy keeps exploring so it does not get stuck in local minima. This is particularly important for our algorithm, considering that the learning of the less important objectives does not start until the important ones are learned.

#### G.3.2 REACHABILITY EXPERIMENT

For Reachability experiment, we use the maze in Figure 12. As we only care about the agent eventually reaching the goal, the agent can completely avoid going on a bad tile. All the policies where it reaches the goal but goes through a bad tile in the process will be dominated by this policy. Hence, we will expect our agent to learn the policy where it eventually reaches the goal and never steps on a bad tile.

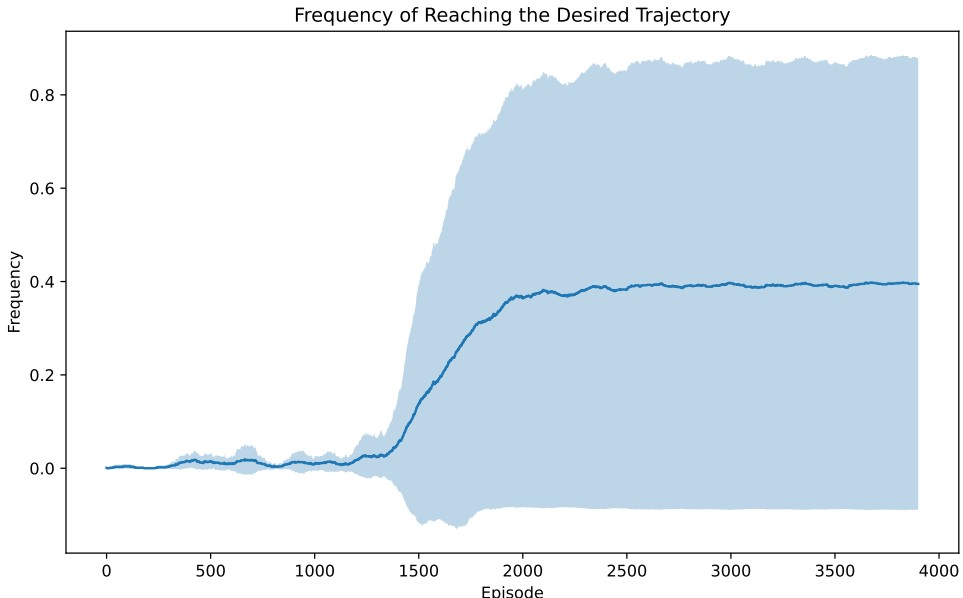

Figure 13: Average satisfaction frequency of 10 seeds for the experiment described in Section G.3.2. The shaded region shows a confidence interval of two standard deviation width around the mean.

We run Algorithm 5 for $N_e = 4000$ episodes and we repeat our experiment with 10 different random seeds. As the policy we use is stochastic, different seeds give significantly different results. Figure 13 summarizes the performance of 10 seeds. The plot shows the ratio of the successful trajectories out of 100 trajectories where successful is defined as satisfying the reachability constraint without stepping on a bad tile. The line shows the mean of 10 different seeds where the shaded region shows the variance in the experiment as two standard deviations around the mean. It can be clearly seen that as the training progresses, the satisfaction frequency increases. Out of the 10 seeds, 4 find policies that have 90% success over 100 episodes.

We can also take a closer look into how the training progresses for a successful seed. Figure 14 shows how the satisfaction frequency for each objective changes throughout the training. It can be seen that the primary objective, reaching the goal eventually starts with a high frequency but drops a little bit while the secondary is being learned. Then, the frequencies for both objectives start to increase together. Intuitively, the initial drop represents when the agent starts to consider "do nothing" policies which reduces the success of the primary objective. But the agent then learns that it can still maintain 0 penalties without just staying in place.

### G.4    ADDITIONAL EXPERIMENTS WITH MORE OBJECTIVES AND ADDITIONAL BASELINES

In this section, we will present additional results using a new benchmark domain from the literature. Fruit Tree Navigation (FTN) Yang et al. (2019) requires the agent to explore a full binary tree of depth $d$ with fruits on the leaf nodes. Each fruit has a randomly assigned vectorial reward $\mathbf{r} \in \mathbb{R}^6$ which encodes the amount of different nutrition components of the fruit. The agent needs to find a path from the root to the fruit with that fits to the user preferences by choosing between left and right subtrees at every non-leaf node.

This domain perfectly highlights the benefits of thresholded lexicographic user preference compared to linear scalarization. The user can have a certain threshold that needs to be reached for each nutrition component and an importance order between these components that should be followed if it is not possible to satisfy all of the thresholds. Using linear scalarization in this case requires knowing the reward values of all of the fruits beforehand, deciding which fruit would fit the user preference, and finding a weight vector $\omega$ for which the desired fruit is better than others.

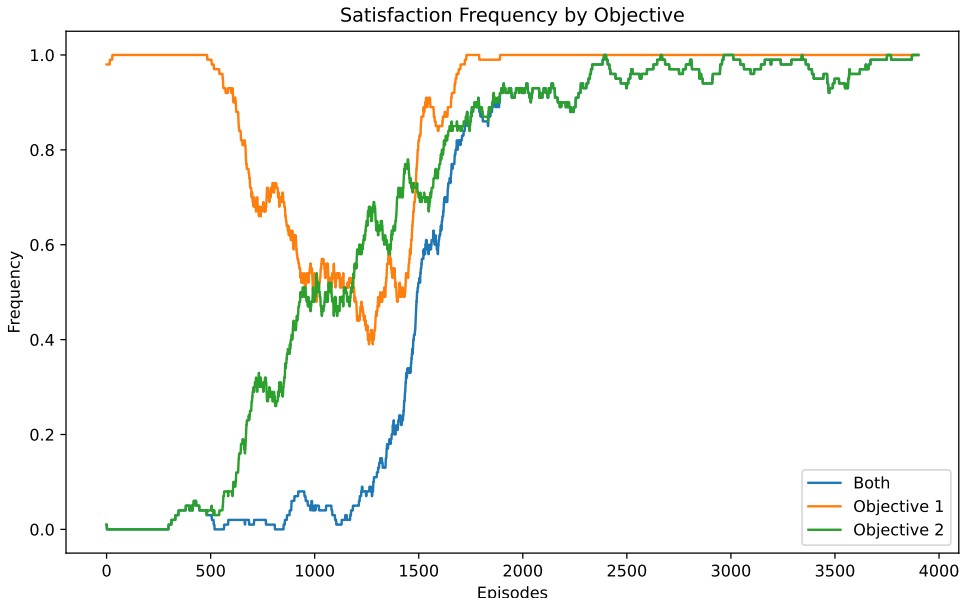

Figure 14: Satisfaction frequency for a single seed for the experiment described in Section G.3.2.

Figure 15 shows the need for the $\Delta$ parameter. It can be seen that using just hyperplanes as done in Uchibe & Doya (2008) fails this task. Similarly, TLQ agent also fails to find the desired leaf.

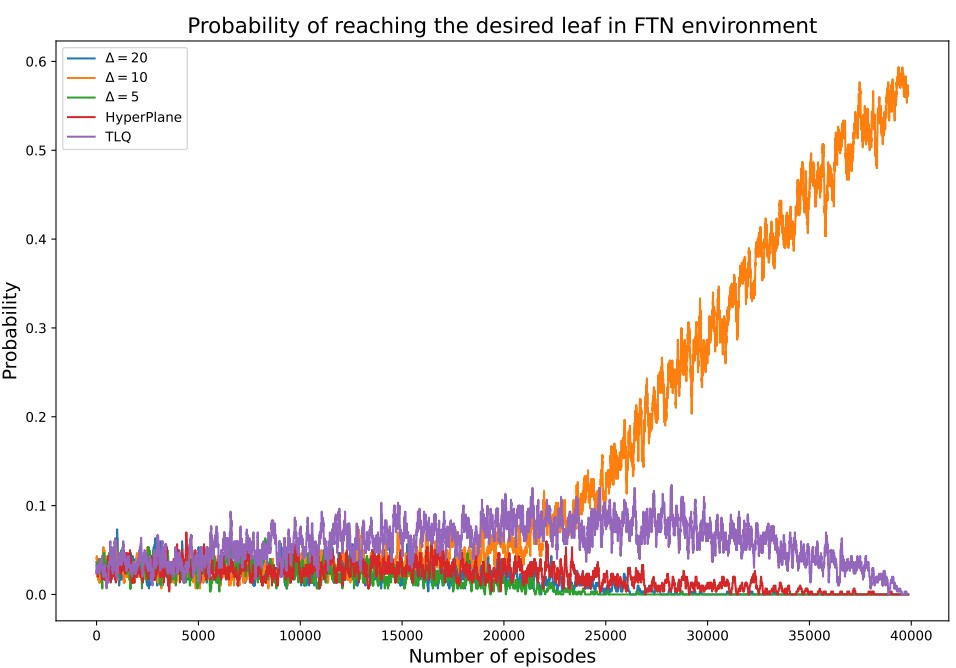

Figure 15: Probability of reaching the desired leaf, averaged over three different random seeds. $\Delta$ values represents conservativeness hyperparameter of the hypercone in degrees.

