# OpenReview forum: "Thresholded Lexicographic Ordered Multi-Objective Reinforcement Learning"
_ICLR.cc/2023/Conference — Submitted to ICLR 2023_

### Official Review · Reviewer_Ekhh · 2022-10-21

**Confidence:** 3
**Correctness:** 2
**Technical Novelty And Significance:** 2
**Empirical Novelty And Significance:** 2
**Recommendation:** 3

**Clarity, Quality, Novelty And Reproducibility:**

The paper is very hard to follow without the appendix. The authors should focus on creating a version of the paper that is more self-contained. The approach of the paper (projecting into hypercones instead of positive half-spaces) is not very novel, the improvements of that in practice are unclear, and there have been investigations into the shortcomings of TLQ. However, the authors present new insights into where these approaches can fail. The experimental evaluation is flawed. Reproducibility is good.

**Strength And Weaknesses:**

Strong points:
* A dive into the shortcomings of TLQ is interesting
* The method could lead to improvements over existing methods. Unfortunately, due to a lack of evaluation, it is unclear how well this method performs.

Weak points:
* The approach and the reasoning why certain things are that way are extremely hard to follow without the appendix. The description of the approach just takes 1 1/2 pages. The authors should write the paper in a more self-contained way.
* The influence and sensitivity wrt. several hyperparameters and how to choose them is unclear (what is a good value for b, ∆,...?)
* The approach of the paper (projecting into hypercones instead of positive half-spaces) is not very novel and the improvements of that in practice are unclear.
* The evaluation of the proposed method has serious flaws:
** The comparison to relevant baselines like TLQ or Lexicographic Multi-Objective Reinforcement Learning (Joar Skalse et al.) is missing. ** All experiments are only done with two objectives. This is problematic because with more objectives finding a good solution is likely much more challenging, and it is unclear how well the method performs with more objectives.
** Comparison to different (existing projections) like into positive half-spaces is missing. Therefore, it is unclear whether the proposed projection into hypercones has benefits in practice.
** It is unclear how the proposed method works in settings where TLQ does not fail.
** Important and commonly used metrics in the evaluation are missing. In RL, the average reward (for each objective) gathered by the policy is one of the most commonly used metrics. However, the authors do not report such metrics.
** The comparison has been made on a single RL setting only; for more benchmarks see for example A Survey on Discrete Multi-Objective Reinforcement Learning Benchmarks, Thomas Cassimon et. al.


**Summary Of The Paper:**

The authors investigate multi-objective RL with thresholded lexicographic ordered objectives. The authors start by investigating the shortcomings of the existing TLQ algorithm. While some of these shortcomings are already known (for example Vamplew et. al.) the authors also show that TLQ does not work under certain circumstances (terminating reachability on the constrained objective and the unconstrained is non-terminating).
After that, the authors propose a lexicographic projection algorithm that projects the gradients onto hypercones and shows how to use the projection algorithm in RL.
In the end, the authors evaluate the lexicographic projection on a simple analytic function. For RL they evaluate their approach on a simple Maze setting only with no comparison to baselines.

**Summary Of The Review:**

Because of the evaluation, the benefits and limitations of the proposed approach in practice are unclear. The experiments done in the paper are insufficient. They are on very few settings (e.g., only two objectives), and relevant baselines, and metrics are not reported. Additionally, the paper should be more self-contained and is very hard to follow without reading the appendix.

---

> ### Author Response · Authors · 2022-11-19
> **Response to Reviewer Ekhh**
>
> Thank you for carefully reviewing the paper; and for the feedback on improving the work.
> Please see below our responses to the review comments.
>
> **The influence and sensitivity wrt. several hyperparameters and how to choose them is unclear (what is a good value for b, ∆,...?)**
>
> Please see Figure 15 of the appendix for sensitivity analysis of $\Delta$ hyperparameter. Our experiments show that the choice of $\Delta$ hyperparameter is important.
>
> **The comparison to relevant baselines like TLQ or Lexicographic Multi-Objective Reinforcement Learning (Joar Skalse et al.) is missing**
>
> We have added TLQ as a baseline in our new experiments with Fruit Tree Navigation in Section G.4 of the appendix.  (Skalse et al.) came out only a few months before our submission and we have not finished adding it as a baseline. It will be added in the camera-ready version.
>
> **All experiments are only done with two objectives. This is problematic because with more objectives finding a good solution is likely much more challenging, and it is unclear how well the method performs with more objectives.**
>
> We added a new benchmark named Fruit Tree Navigation which has 6 objectives. Our experiments show that our algorithm scales well to more objectives. Please see Section G.4 and Figure 15 of the appendix.
>
> **Comparison to different (existing projections) like into positive half-spaces is missing. Therefore, it is unclear whether the proposed projection into hypercones has benefits in practice.**
>
> We added halfspace projection as a baseline in our new experiments in Section G.4. This baseline can be seen in Figure 15 under the name HyperPlane. The experiment indicates the need for hypercone projection.
>
>
> **It is unclear how the proposed method works in settings where TLQ does not fail.**
>
> This is an interesting direction for our future work. Thanks for the suggestion.
>
> **Important and commonly used metrics in the evaluation are missing. In RL, the average reward (for each objective) gathered by the policy is one of the most commonly used metrics. However, the authors do not report such metrics.**
>
> Since the goal in our setting is finding a specific Pareto optimal policy, we believe metrics like average reward is not as informative as the success rate of the agent in reaching that policy.
>
> **The approach of the paper (projecting into hypercones instead of positive half-spaces) is not very novel and the improvements of that in practice are unclear.**
>
> While the positive half-space projection approach is similar to our technique, our setting has important differences like an importance ordering. As illustrated in Figure 15 of the appendix, using hypercones instead of positive half-spaces leads to significant improvements.

---

### Official Review · Reviewer_oJxc · 2022-10-23

**Confidence:** 5
**Correctness:** 4
**Technical Novelty And Significance:** 2
**Empirical Novelty And Significance:** 1
**Recommendation:** 3

**Clarity, Quality, Novelty And Reproducibility:**

The paper is easy to understand and it is well written.
Reproducibility is questionable but in view of the weakness of the experiments this is a no issue.

**Strength And Weaknesses:**

Strengths:
The idea of using PG for MO-RL.
The example showing the drawbacks for the thresholding algorithms.

Weaknesses:
Some references and algorithms are missing in the background work (and experiments). See M. Fleischer. The measure of Pareto optima applications to multi-objective metaheuristics. EMO, Springer Verlag, pages 519-533, 2003.
The computational experiments are not strong. It seems the experiments only deal with the proposed algorithm and don't compare the algorithm vs existing algorithms.

**Summary Of The Paper:**

Multiobjective (MO) RL  has many applications with the thresholding algorithm often used. The authors point out a deficiency of such algorithms and propose a PG based approach. They also conduct a numerical study comparing their algorithm with benchmark algorithms.

**Summary Of The Review:**

While adapting PG to MO is new the underlying ideas are not significantly innovative.
I don't find the identified problems with reachability of significant importance.

Lack of sound experiments is a significant drawback of the work. There are no benchmark algorithms and the datasets are very easy.

There is also abundant work in MO-MAB that might be applicable here. While MAB doesn't have the notion of PG, they can serve as benchmarks.

---

> ### Author Response · Authors · 2022-11-19
> **Response to Reviewer oJxc**
>
> Thank you for carefully reviewing the paper; and for the feedback on improving the work.
> Please see below our responses to the review comments.
>
> **Some references and algorithms are missing in the background work.**
>
> We will add the suggested work in Related Work.
>
> **Lack of sound experiments is a significant drawback of the work. There are no benchmark algorithms and the datasets are very easy.**
>
> To address these concerns, we tested our algorithm on a new benchmark and compared it with additional baselines. Fruit Tree Navigation is a 6 objective task that requires the agent to navigate through a full binary tree to find the fruit that fits the user preferences. We also compared our results with TLQ and an agent that uses hyperplane projection instead of hypercone. Please see Section G.4 and Figure 15 of the appendix for more details.
>
> **There is also abundant work in MO-MAB that might be applicable here. While MAB doesn't have the notion of PG, they can serve as benchmarks.**
>
> We are not aware of any work that studies the applicability MO-MAB techniques to control tasks. However, we believe that this can be an interesting direction for future work.

---

### Official Review · Reviewer_Jg7J · 2022-10-25

**Confidence:** 4
**Correctness:** 2
**Technical Novelty And Significance:** 3
**Empirical Novelty And Significance:** 2
**Recommendation:** 3

**Clarity, Quality, Novelty And Reproducibility:**

**Clarity:** This paper is well organized and can be easily understood.

**Quality:** This work is not complete, since it needs more theoretical proof and experimental verification.

**Novelty:** This paper introduces a new algorithm, but the motivation of this work is not well-supported.

**Reproducibility:** This work can be reproduced.

**Strength And Weaknesses:**

**Strengths**:

1. The problem this paper tries to solve is greatly concerned in the RL community.
2. The idea of solving lexicographic MORL problems with policy-based algorithms is a good attempt. The LPA algorithm is very versatile since it can be combined with any gradient-optimization algorithms and any policy-gradient (and actor-critic) algorithms.
3. This paper is well organized.

**Weaknesses**:

1. The motivation is not well-supported. This paper focus on the lexicographic multi-objective problems that can not be solved by Lexicographic Q-Learning (TLQ), and two types of maze example are analyzed. Each of them will be discussed below.
   1) Problems with Reachability Constraint (Fig.1). The problem can be solved by TQL, but the problem itself needs to be redefined. The problem should contain a third objective time cost. The primary objective (Reach G) does not consider the time spent, i.e. no discounting situation, so the algorithm converges to a random policy (described in the paper as "all actions would have the value $\tau_1$"). If the time factor is not taken into account, the random policy is right, but it's not an ideal policy. The discount factor is a mathematical trick to make an infinite sum finite, and it also chooses a suitable planning horizon. So discount factor can be viewed as a trade-off of task rewards and time costs, which is also necessary in reality, because no one can do something without considering the time. Then, the secondary objective (avoid bad tiles) will lead to another stochastic policy that does not step on tiles but may move back and forth. The third objective (time cost) can be formalized as a time penalty or discount factor, leading to the deterministic ideal policy.
   2) Problems with Non-reachability Constraint (Fig.3). This problem (or just the example in Fig.3.) can be solved by TQL. The primary objective (minimizing the cost of tiles) can converge to the unique path (including *(0-3,0), (3,1), (0-3,2), (0,3), (0-1,3)* ), but there may be backtracking (like *Left* in *(1,0)* with previous step *Right* in *(0,0)*). And the secondary objective (minimizing the time) then allows the agent to reach the goal without backtracking, i.e., in 11 steps.
2. If the number of objectives is greater than or equal to 3, which means that "return None" of Algorithm 2 may occur, no theoretical proof of the convergence or experimental verification is given in this paper.
3. Here are some minor typographical errors and suggestions.
   1) Spelling error on page 4, line 9. "iff" $\rightarrow$ "if"
   2) The full name of LMDP should be given before using LMDP in Sec. 1.
   3) The description of Fig. 1 uses $H$s and $h$s indicate tiles, while $HH$s and $hh$s are in the description of Fig. 3.

**Summary Of The Paper:**

This paper focuses on lexicographic multi-objective problems. Firstly, the shortcomings of the existing algorithm Lexicographic Q-Learning (TLQ) are analyzed, and the scenarios in which it is not applicable are pointed out. Secondly, this paper proposes the lexicographic projection algorithm (LPA), which performs multi-objective optimization by a heuristic projection of gradients, so that the current optimization targets the highest priority objective that does not reach the threshold while preventing the degradation of higher priority objectives as much as possible. Finally, this paper combines LPA with a policy-based reinforcement learning algorithm and validates it in the MAZE environment.

**Summary Of The Review:**

I think this paper introduces a new algorithm combined with policy gradient, which approaches the lexicographic multi-objective problem from a new perspective. However, this paper has some fatal flaws. (1) The motivation is not well-supported. (2) The experiment needs to be more fully validated.

---

> ### Author Response · Authors · 2022-11-19
> **Response to Reviewer Jg7J**
>
> ### Problems with Reachability Constraint (Fig.1):
>
> **Time through discounting**
>
> Including time as discounting does not work as discussed in Section 4 and Appendix C.1. Discounting with Absolute Thresholding causes all state-action pairs except the ones that immediately lead to $G$ to have values less than $\tau_1$ w.r.t. the primary objective. This means that only the primary objective will be considered. Hence, the policy learned in this setting will result in the trajectory $(1,0) \to (1,1) \to (2,1)$. Seeing how the other methods fail requires a closer examination of the value function, please refer to Appendix C.1 to see our explanation. As a result, it can be seen that this task is not solvable with any of these TLQ variants with or without discounting when it is defined as a two-objective task.
>
> **Time as a third objective**
>
> Redefining the task to include time as a third objective significantly alters the nature of the task. Firstly, as stated in "TLQ does not work" part of our summary of the applicability of TLQ in Section 4, we show that TLQ fails when the constrained objective is a terminating reachability objective but the unconstrained objective is non-terminating. However, since time-cost is a terminating objective, the task would not fall under this category if time is added as the third objective.
>
> Secondly, the tile-cost objective in the resulting task will be a constrained objective since a third objective is added. Since tile-cost objective is a non-reachability objective, this would fall under the category shown to be failing in (Vamplew et. al. 2011). The main reason for this failure is the fact that the agent cannot keep track of the cost it already incurred.
>
> Please note that this specific instance might be in fact solvable by TLQ, as this is a special case where the allowed cost would be 0. In this case, the agent does not need to keep track of the incurred cost as under the optimal policy it never incurs a tile cost. However, this is a special case that does not reflect the requirements in the general case.
>
> ### Problems with Non-reachability Constraint (Fig.3).
>
> The claim that TLQ fails on this type of problems in general is failing comes from (Vamplew et. al. 2011). Also, the path we are looking for is (0-3,0), (3, 1-4), (2,4), (1,4) which represents a Pareto optimal compromise between two objectives. The path suggested by the reviewer is different than ours. This might be achievable using TLQ as it is a special case where accounting for the tile cost so far is not needed as the threshold can be set to 0.
>
> This is a good example to highlight again that all existing analyses of TLQ, including our work, are done empirically for the cases both where it works and where it does not. While the empirical results are supported by intuition, there are not any theoretical guarantees. Therefore, there may be corner/special cases where the analysis does not hold.
>
> ### When the number of objectives is greater than 2
>
> We have added the new results with Fruit Tree Navigation (FTN) domain to experimentally verify our algorithm. FTN has 6 objectives and our experiments show that our algorithm can successfully solve it while baselines like half-space projection and TLQ fail it. Please see Section G.4 and Figure 15 of the appendix.
>
> ### Here are some minor typographical errors and suggestions.
>
> We’ve fixed this in the new revision.

---

### Official Review · Reviewer_NWsv · 2022-11-02

**Confidence:** 3
**Correctness:** 3
**Technical Novelty And Significance:** 3
**Empirical Novelty And Significance:** 3
**Recommendation:** 6

**Clarity, Quality, Novelty And Reproducibility:**

Clarity and quality: Both extremely high
Novelty: Methods in this space are sparse, as the authors discuss, and the authors approach is fairly novel (as a synthesis of several existing ideas in the literature).
Reproducibility: The authors provide some code, and promise to release more.

**Strength And Weaknesses:**

Strengths: The paper did an excellent job formalizing the problems existing with current methods (especially failure cases), as well as how the author's approach directly addresses these problems.

Weaknesses: It's a tired objection, but it's somewhat difficult to situate the author's results with respect to the rest of the literature.  While this is certainly true of any novel work, for which there isn't much else to compare to, then the onus is exceptionally upon the authors to provide compelling results on difficult problems.  While the problems that the authors chose to present were great as educational devices demonstrating the utility of their methods, it's unclear how _practically_ useful the author's algorithm is.  It would be great if the authors could further discuss how their algorithm might fare in a more difficult, less synthetic setting.

**Summary Of The Paper:**

The authors discuss shortcomings of existing approaches for thresholded lexicographic ordered multiobijective problems in reinforcement learning.  Additionally, the authors provide a policy gradient algorithm that performs well on this class of problems.

**Summary Of The Review:**

A solid submission tackling a fairly new space.  The manuscript could use more extensive, and more practically relevant experiments, though.

---

> ### Author Response · Authors · 2022-11-19
> **Response to Reviewer NWsv**
>
> Thank you for carefully reviewing the paper; and for the feedback on improving the work.
>
> We added experiments with a new benchmark from the literature which has more objectives and larger state space. Please refer to Section G.4 and Figure 15 of the appendix for these new experiments. We agree that more experiments on harder benchmarks are needed to further validate our approach. We consider our work as a first step in using policy gradient methods in this setting which should be supported by future work.

---

### Decision · Program_Chairs · 2023-01-20

**Decision:**

Reject

**Justification For Why Not Higher Score:**

The reviewers mostly agree that the paper should be rejected, and I also agree with them.

**Justification For Why Not Lower Score:**

N/A

**Metareview: Summary, Strengths And Weaknesses:**

The paper looks at lexicographic multi-objective RL problems, identifies some issues with existing methods and recommends some fixes in the algorithms. The approach is experimental only, and the experiments are not convincing enough (e.g., the algorithm is extremely sensitive to the parameter $\Delta$, thus it is not clear that the improvement is achievable) -- the authors also promise experiments for future work, which they should indeed do if they intend to resubmit the paper to another venue. The reviewers point out several issues.

An additional comment not raised in the reviews: By definition, a reachability objective has 0 reward if the episode does not terminate within the time horizon, and positive reward if it does. Thus, by definition, it is always a terminating objective. Given this, it is not clear what the authors mean by a non-terminating reachability objective (the authors wrote "constraint" in the paper, but I assume it should have been "objective").